

# Streamflow characteristics from modelled runoff time series - importance of calibration criteria selection

Sandra Pool[1], Marc J. P. Vis[1], Rodney R. Knight[2], and Jan Seibert[1,3,4]

[1]Department of Geography, University of Zurich, Zurich, Switzerland
[2]U.S. Geological Survey Lower Mississippi—Gulf Water Science Center, 640 Grassmere Park, Suite 100, Nashville, TN 37211, USA
[3]Department of Earth Sciences, Uppsala University, Uppsala, Sweden
[4]Department of Physical Geography, Stockholm University, Stockholm, Sweden

*Correspondence to*: Sandra Pool (sandra.pool@geo.uzh.ch)

**Abstract.** Ecologically relevant streamflow characteristics (SFCs) of ungauged catchments are often estimated from simulated runoff of hydrologic models. Estimated SFCs can be substantially uncertain when models are calibrated using traditional approaches based on minimization or maximization of statistical performance metrics (e.g. Nash–Sutcliffe efficiency). To evaluate model performance, we tested how well SFCs are simulated when the model objective function was calibrated using one or more SFCs. We calibrated a bucket-type runoff model for 25 catchments in the Tennessee River basin and evaluated the proposed calibration approach on 13 selected SFCs representing major flow regime components and different flow conditions. While the model tends to underestimate SFCs related to mean and high-flow conditions, SFCs related to low flow are overestimated. The highest estimation accuracies were achieved by a SFC-specific model calibration. Estimates of SFCs not included in the calibration process were of similar quality when comparing a multi-SFC calibration approach to a traditional Nash–Sutcliffe efficiency calibration. For practical applications, this implies that SFCs should preferably be estimated from targeted runoff model calibration and modelled estimates need to be carefully interpreted.

## 1    Introduction

Reliable runoff information is fundamental for many water resources-related tasks such as flood prevention, drought mitigation, management of drinking water supply and hydropower, or river restoration. Runoff modelling is a tool that can be used to create runoff time series when observed time series are not available. Runoff model simulations usually focus on accurately simulating specific runoff characteristics relevant to a respective application. The extraction of runoff characteristics from a simulated time series may produce poor estimates when these characteristics were not included in model calibration. A typical example is the use of runoff simulations for the estimation of streamflow characteristics (SFCs). SFCs are properties of the annual streamflow hydrograph defining the structure and functioning of aquatic and riparian biodiversity (Richter et al., 1996; Poff et al., 1997). The accurate prediction of streamflow characteristics is a core determinate to defining how streamflow and aquatic communities relate. A large number of SFCs have been suggested to



characterize ecologically relevant aspects of the flow regime (Tharme, 2003) and have become the basis for decision-support systems integrating resource management with ecological response.

Multivariate regression or runoff models are used to estimate SFCs when observed streamflow time series data are not available (Hailegeorgis and Alfredsen, 2016). The estimation of SFCs with linear regression usually relates a single SFC to

catchment characteristics such as climate, land cover, geographic, and geologic variables (e.g. Sanborn and Bledsoe, 2006; Carlisle et al., 2010; Knight et al., 2012). This approach is inflexible in a sense that the regression is SFC-specific and does not allow for analysis of potential water-use and land management (Murphy et al., 2013). These disadvantages can be partially overcome by applying runoff models. Simulated streamflow time series from runoff models can be used to calculate any SFC and by changing model input and parameters different scenarios  such as climate change, groundwater withdrawals,

land use and riverine change can be simulated (Poff et al., 2010; Murphy et al., 2013; Olsen et al., 2013; Shrestha et al., 2014).  While runoff models provide flexibility in evaluating scenarios, statistical models such as multiple linear regressions often provide greater accuracy (Murphy et al., 2013).

Runoff models are used in both ecohydrology and hydrological modelling as tools to simulate specific aspects of the runoff regime. The terms, SFCs or ecological flow indices, are often used to refer to such specific aspects of the flow regime in

ecohydrology studies, whereas the more recently introduced term, hydrological signatures, has been used in hydrological modelling (Jothityangkoon et al., 2001; Wagener et al, 2007). Hydrological signatures can often support a physical interpretation of the way a catchment functions and are seen as valuable metrics especially for modelling ungauged catchments (Jothityangkoon et al., 2001), for selecting appropriate model structures (Euser et al., 2013), and for classifying catchments (Wagener et al., 2007; Sawicz et al., 2011). Regardless of the terminology and the ultimate goal, the basic goal is

the quantification of certain aspects of a streamflow time series to answer various questions such as the response of aquatic health to changes in a flow regime. In this paper, we use the term SFC as equivalent to hydrological signature.

Estimated streamflow characteristics are prone to significant errors when calculated from simulated time series (Murphy et al., 2013; Shrestha et al., 2014; Vis et al., 2015). This is due in part to the objective functions used for evaluating the model error such as the commonly used Nash–Sutcliffe efficiency (Nash and Sutcliffe, 1970) or volume error, which do not ensure

that a model is reproducing particular streamflow characteristics. These objective functions subsequently guide model parameter calibration, which strongly influences the simulated hydrograph (for an overview see Pfannerstill et al., 2014) in terms of annual, seasonal, and monthly volumes and magnitudes. The large variability in estimated SFC accuracy as well as the bias in the estimates can be observed independent of the model used to simulate the runoff time series (Caldwell et al., 2015). A remedy to this large variability and bias is to incorporate SFCs into model calibration schemes. For example,

Westerberg et al. (2011) and Pfannerstill et al. (2014) focused on specific evaluation points or segments of the flow-duration curve (FDC) during model calibration. Both studies report better overall performance for the simulated hydrograph with a FDC-based calibration compared to a more traditional calibration approach using, for example, the Nash–Sutcliffe efficiency (Nash and Sutcliffe, 1970). However, runoff models calibrated using FDC have to be constrained by additional SFCs if one is interested in more esoteric and subtle aspects of the flow regime such as the timing of events or snow-related runoff





processes (Westerberg et al., 2011). Instead of aiming at a well-simulated, general hydrograph, Hingray et al. (2010) and Olsen et al. (2013) focused on certain aspects of the streamflow regime that were considered most important. Their results, which are echoed by Murphy et al. (2013), suggest that the runoff model performs reasonably well for the aspects on which it is calibrated, whereas it only modestly represents other runoff characteristics. Hence, developing an approach to increase

the accuracy of estimated SFCs from runoff model time series continues to be an open challenge in hydrological modelling.

The main objective of this study was to assess the potential for a runoff model calibrated using specific aspects of the flow regime to more accurately estimate a suite of SFCs as compared to using more traditional calibration approaches. The general approach was based on the idea that most information essential for estimating SFCs is preserved in the simulated hydrograph by including selected SFCs in model calibration. Our modelling approach relies on catchments with observed

runoff time series and therefore does not answer the question of how to simulate SFCs in ungauged or altered catchments. However, the prediction of runoff for ungauged catchments benefits from an improved and informed calibration strategy for gauged catchments, which is used in the subsequent regionalisation. For regionalization approaches we refer to studies such as Yadav et al. (2007), Viglione et al. (2013) or Westerberg et al. (2016).

The following questions are addressed in this paper:

(1)  How well is a single SFC simulated when that SFC is used in the model objective function?

    (2)  How well is a single SFC simulated when the model objective function contains one or more other SFCs?

    (3)  How does the accuracy of estimated SFCs vary between traditional calibration approaches and those where specific SFCs are included?

## 2    Methods

### 2.1    Catchment locations and characteristics

The study catchments are all located in the 106000 km$^2$ Tennessee River basin in the southeastern United States (Fig. 1), which is one of the most diverse temperate freshwater ecosystems in the world (Abell et al., 2000). A large number of endemic fish species and a unique assemblage of mussels, crayfish and salamanders make the Tennessee River basin an excellent area for ecohydrological studies (Abell et al., 2000). From a study published by Knight et al. (2008), 25 catchments

in the Tennessee River basin having observed streamflow time series (U.S. Geological Survey, 2016b), precipitation (U.S. Department of Commerce, 2007a), temperature (U.S. Department of Commerce, 2007b) and potential evaporation data (Rotstayn et al., 2006) were selected. The sizes of catchment areas range between 100 and 4800 km$^2$ with elevations ranging from 174 to 937 m (U.S. Geological Survey, 2016a) above the North American Vertical Datum of 1988 (NAVD 88). Land cover for the study catchments is predominantly hardwood forest and pasture. Air temperature and precipitation varies

between catchments according to both catchment elevation and longitude. Mean annual air temperature in the 25 catchments varies between 9.3 and 14.7° C, and annual precipitation varies from 1500 to 2020 mm with autumn being slightly drier and less than 8% of annual precipitation falling as snow. Runoff is highest in winter and lowest in summer, ranging from 400 to



1300 mm a$^{-1}$ (millimeters per year). Variability in soil thickness (Omernik, 1987), regolith thickness, karst development and topographic slope (Hoos, 1990; Wolfe et al., 1997; Law et al., 2009) are documented as asserting the most influence on runoff.

## 2.2 Selection of SFCs

Thirteen SFCs assessed in this study were chosen for use in model scenarios based on discernible functional connections with fish community diversity (Knight et al., 2008; Knight et al., 2014). This set of 13 SFCs represents each of the major flow regime components commonly used in ecological studies (e.g. Olden and Poff, 2003; Arthington et al., 2006; Caldwell et al., 2015): magnitude, ratio, frequency, variability and date (Table 1). For this study the SFCs were additionally grouped according to flow conditions (mean, low and high flow), because different aspects of the hydrograph have been shown to be
sensitive to the objective function used for model calibration (for an overview see Pfannerstill et al., 2014).

## 2.3 The runoff model

The HBV (Hydrologiska Byråns Vattenavdelning) model (Bergström, 1976; Lindström et al., 1997) is a bucket-type hydrologic model for simulating continuous runoff series. Model inputs are daily rainfall and air temperature, as well as daily potential evaporation values. Hydrologic processes are represented by four different routines corresponding to snow,
soil water, groundwater, and runoff routing, with a combined total of 16 parameters. In the snow routine, snow accumulation and snowmelt are calculated by a degree-day method. Snowmelt together with rainfall and potential evaporation are input to the soil-water routine, where the actual evaporation and the groundwater recharge are computed based on the soil-moisture storage. The groundwater (or response) routine consists of a connected shallow and deep groundwater reservoir and simulates peak flow, intermediate runoff and baseflow. These three runoff components are taken together and transformed
by a triangular weighting function during the routing process to calculate the runoff at the catchment outlet. Runoff can be modelled in a semi-distributed way by separating a catchment into elevation bands. Thereby, the snow and soil-water routines are calculated for each elevation band, whereas the groundwater storage and the runoff routing routines are treated as a lumped representation of the entire catchment. HBV exists in different versions, whereby the general structure of the model remains the same. The version applied in this study is HBV-light (Seibert and Vis 2012). Like for all bucket-type
models, parameters in the HBV model cannot be determined *a priori*, they are identified by model calibration instead. More detailed information on the HBV model can be found in Bergström (1976), Lindström et al. (1997) and Seibert and Vis (2012).





### 2.4 Modelling approach

#### 2.4.1 Model setup

For each of the 25 catchments the number of elevation bands was defined by splitting the catchment into elevation zones of 200 m. Elevation zones covering less than 5% of the catchment area were merged with the adjacent elevation zone. For the
resulting elevation bands, air temperature and rainfall were computed with a lapse rate of 6° C per 100 m and 10% per 100 m, respectively. Potential evaporation was assumed to be uniform over the whole catchment.

Model simulations were run for two time periods, one lasting from the hydrological years (1st of October until 30th of September) 1984 to 1996 and the other lasting from 1997 to 2009. The approximately three years preceding each simulation period served to establish state variables of the model. A three-year calibration period was needed to ensure that the different
state variables at the beginning of the simulation period were consistent with the preceding meteorological conditions and parameter values. The two simulation periods were used for model calibration and validation. For calibration, a genetic algorithm (Seibert, 2000) was used and the range of possible parameter values was specified based on previous studies (Lindström et al., 1997; Seibert, 1999; Table 2 in Vis et al., 2015). The 100 independent calibration trials allowed to account for parameter uncertainty or equifinality (Beven and Freer, 2001) and resulted in a set of 100 calibrated parameter sets for
each objective function (Fig. 2).

#### 2.4.2 Choice of objective functions for model calibration

The complete model calibration process was conducted for 25 catchments and using data from all five different types of objective functions (see Table 2 for the exact equations) that focused on different aspects of the hydrograph. In the first step, model parameters were constrained maximizing the Nash–Sutcliffe efficiency criterion ($R_{eff}$, Nash and Sutcliffe, 1970). The
Nash–Sutcliffe efficiency is the most widely used objective function in hydrological modelling, and it served as a benchmark for the objective functions that included SFCs. Model calibration with $R_{eff}$ tends to reduce simulation errors in magnitude and timing of high-flow conditions at the expense of errors in low-flow conditions (Legates and McCabe, 1999; Krause et al., 2005).

Next, a new efficiency measure that consisted of one single SFC ($I_{Single}$) was defined to explicitly incorporate each of the 13
SFCs (Table 2). Additionally, each SFC efficiency measure was combined with $R_{eff}$, whereby both metrics were equally weighted ($I_{Single\_Reff}$). The use of a single SFC as the objective function allowed calibration to focus on a specific aspect of the hydrograph, while adding $R_{eff}$ helped to improve the overall shape of the hydrograph including the magnitude and timing of events.

Based on the results from the individual SFCs, an objective function consisting of four different and equally weighted SFCs
was defined ($I_{Multi}$, Table 2). This SFC based efficiency measure was again combined with $R_{eff}$ ($I_{Multi\_Reff}$). For the resulting combined objective function, weights of 0.2 were assigned to each metric to make sure the individual SFCs had sufficient influence on the model calibration and were not dominated by $R_{eff}$. The number of SFCs constituting $I_{Multi}$ was not previously





fixed. Instead, a minimum number of SFCs was defined so that the objective function was both robust and informative. A SFC was considered as robust when the SFC calculated from a model simulation with $I_{Single}$ had small errors over the full range of catchments in both validation time periods. A SFC was regarded as being informative, when it also yielded relatively good simulations for other SFCs.

### 2.4.3 Evaluation of model performance

Model performance in calibration and validation was evaluated by means of SFCs, $R_{eff}$ and mean absolute relative error (MARE) (see Table 3 for the exact equations). These evaluation criteria were calculated for all 100 runoff simulations based on the five different types of objective functions in both validation time periods and for all 25 catchments. For the interpretation of the results, the median parameter set of each catchment was selected.

The SFCs were calculated using the U.S. Geological Survey (2014) EflowStats R-package. As there are significant differences in the SFC ranges, a normalization was needed that allowed comparison of the different SFCs. Instead of normalizing in terms of relative error, an approach was applied that normalizes the SFC estimation error. The normalization of a SFC was computed as the absolute simulation error divided by the range of possible values for that SFC in the respective catchment (Table 3). To calculate these SFC ranges, 10000 Monte Carlo simulations were run for each respective catchment using randomly chosen parameter values from the previously identified parameter space (Lindström et al., 1997; Seibert, 1999; Table 2 in Vis et al., 2015). The Monte Carlo simulations represented the potential variation in a certain SFC if no information was available to constrain the runoff model. The range was then calculated as the difference between the 10[th] and 90[th] percentiles of the simulated SFC values.

## 3 Results

### 3.1 The use of single SFCs as objective functions in model calibration

### 3.1.1 How informative is a SFC for estimating any SFC?

The calibrations for all 13 versions of $I_{Single}$ and $I_{Single\_Reff}$ resulted in 13 different runoff simulations that were evaluated by calculating the normalized SFCs for the calibration and validation periods. The SFC TA1 (stability of runoff) was selected as a representative example to illustrate that model calibration with $I_{Single}$ resulted in greater variability in model performance than the calibrations with either $I_{Single\_Reff}$ or $R_{eff}$, independent of the considered time period (Fig. 3, where the spread along the $I_{Single}$-axis is larger than the spread along the $I_{Single\_Reff}$ or $R_{eff}$-axis). While estimation accuracies with $I_{Single\_Reff}$ and $R_{eff}$ are often of comparable magnitude, they both outperform most simulations with $I_{Single}$. Error magnitudes from the three described objective function types ($I_{Single}$, $I_{Single\_Reff}$ and $R_{eff}$) can vary considerably between time periods (triangles and circles respectively in Fig. 3).





The median simulation error of all 13 versions for each objective function ($I_{Single}$ and $I_{Single\_Reff}$) with each SFC is presented in Fig. 4. The use of SFC as a single objective function ($I_{Single}$) generally resulted in poor SFC estimations for those SFCs not included in $I_{Single}$ in both the model calibration and validation. The SFC estimates became substantially better, with narrow spread and lower median of the absolute normalized SFC error, when $I_{Single}$ was combined with $R_{eff}$.

### 3.1.2    Estimation accuracy using SFC-specific model calibrations

Model calibration results for the 13 SFCs confirmed that HBV is capable of estimating different SFCs with a high level of precision if the respective SFC is used as an objective function ($I_{Single}$) for model calibration (Fig. 5a). Using the combined objective function $I_{Single\_Reff}$ gave similar, although slightly less precise results, whereas calibrations using $R_{eff}$ as the objective function resulted in the least accurate estimates. However, $I_{Single}$ yielded poor model performances in relation to $R_{eff}$ if $R_{eff}$ was not combined with the objective function $I_{Single}$.

Validation results exhibited a similar pattern in model performance (Fig. 5b). The median absolute normalized error of the 13 SFCs was relatively low for model runs based on the objective functions $I_{Single}$ and $I_{Single\_Reff}$ and was higher for simulations based on the model calibration with $R_{eff}$. The inclusion of $R_{eff}$ into the objective function had a negative effect on the model performance, especially for FL2 and MA26 (Fig. 6a-c). Except for MH10, which was best estimated with the objective function $R_{eff}$, SFCs can be regarded as valuable for model calibration.

### 3.2    The use of multiple SFCs for model calibration

Figure 7a shows simulation results for the objective function $I_{Single}$ for all 25 catchments and both modelling time periods. The five SFCs with the highest robustness (less variability in error; Fig. 7a) were RA7, ML20, FH6, E85 and MA41. The information value of these five SFCs varied, but all together each of the 13 SFCs were well simulated by at least one of these five (Fig. 7b). However, since the information value of ML20 (base flow) and E85 (lowest 15% of daily runoff) was redundant, E85 was discarded as a potential SFC for the objective function $I_{Multi}$.

Median estimates of the 13 SFCs in the calibration period were slightly lower when the model was calibrated with $I_{Multi}$ rather than $I_{Multi\_Reff}$. Both of these objective functions led to much better model performance for SFCs than calibrating with $R_{eff}$ alone. The inverse pattern was observed when evaluating model performance in terms of $R_{eff}$ and MARE (Fig. 5a).

Model performance for the validation period with $I_{Multi\_Reff}$ had lower median error than the error associated with using $I_{Multi}$ as objective function (Fig. 5b). The comparison of $I_{Multi}$ and $I_{Multi\_Reff}$ for all SFCs separately confirmed the small differences by showing that for most SFCs both objective functions resulted in similar estimates (Fig. 8a). While the two objective functions had a comparable performance in terms of SFC, the result diverged when evaluating their efficiency for $R_{eff}$ and MARE. The two criteria, $R_{eff}$ and MARE, were better simulated with $R_{eff}$ being part of the objective function (Fig. 5b).

As could be expected, there was a pronounced difference in median estimates of SFCs between model simulations with the objective functions $I_{Multi\_Reff}$ and $I_{Single\_Reff}$. $I_{Single\_Reff}$ was clearly better for estimating SFCs, especially for SFCs not included in the $I_{Multi\_Reff}$ objective function (Fig. 8b). Comparing simulations from $I_{Multi\_Reff}$ and $R_{eff}$ revealed a smaller median error of



the SFCs (Fig. 8c) but poorer efficiencies for $R_{eff}$ and MARE when calibrating with $I_{Multi}$ (Fig. 5b). Yet, for most SFCs not explicitly incorporated into the objective function $I_{Multi\_Reff}$, $R_{eff}$ performed equally well or slightly better than $I_{Multi\_Reff}$ (Fig. 5b).

### 3.3    Estimation accuracy for SFCs

Figure 9 provides an overview of how well SFCs were simulated by presenting the results for both modelling time periods and all five objective function types. Performance values were categorized as small (< 10%), medium (11–20%), large (21–30%) and very large (>30%) errors. The median error was used for the evaluation of the under- or overestimation. An underestimation of SFC values was observed for all five SFCs representing high-flow conditions as well as for three of four mean-flow related SFCs. With one exception, low-flow SFCs were overestimated. The magnitude of the absolute error varied from generally small for RA7, ML20, MH10 and FH6, to medium for MA41, TA1 and DH16, and up to very large magnitude for TL1. A considerable range, from small to large errors, was observed in the individual objective functions for FL2, MA26, E85, MH10, DH13, FH7, and TL1. Except for four SFCs, the magnitude of the simulation error depended either on the time period (MA26, E85, TL1, DH13, DH16) or the objective function (RA7, MH10, FH6, FH7) considered. These groups of SFCs regarding magnitude, spread and dependence of the error did not seem to be related to the flow components (magnitude, ratio, frequency, variability and date) or flow conditions (low, medium and high flow).

Normalized errors for the high-flow conditions, DH16 and MH10, for all 25 catchments and for both modelling time periods indicate two typically observed phenomena regarding uncertainty due to differences in catchments. DH16 is an example of a SFC that could be regarded as being clearly underestimated by the model, because of its negative bias in nine out of ten cases (Fig. 10a). However, for objective functions or modelling time periods with a low magnitude in the median bias, there might be a substantial number of catchments that show overestimation of DH16. A second commonly observed phenomenon is shown by the SFC MH10 (Fig. 10b). While MH10 had mostly small median errors, there were many catchments with considerably higher errors. Although MH10 was the most extreme example, it illustrates that small median errors do not guarantee good results for all catchments.

## 4    Discussion

### 4.1    On the importance of the choice of the objective function

The results demonstrated that the objective function used for model calibration strongly influences the estimation accuracy of SFCs. This finding confirms the findings of previous studies (e.g. Hingray et al., 2010; Westerberg et al., 2011; Murphy et al., 2013; Olsen et al., 2013; Pfannerstill et al., 2014; Shrestha et al., 2014; Caldwell et al., 2015; Vis et al., 2015) and points out the importance of making a careful choice of the objective function for model calibration. As can be expected, a particular SFC is best estimated when the model calibration is based on that SFC ($I_{Single}$). However, a SFC-specific model calibration generally results in rather poorly simulated hydrographs, which negatively affects the estimation accuracy of



SFCs that were not included in the model calibration. This poor estimation of SFCs can be improved by constraining model parameters not only to one SFC but also to $R_{\text{eff}}$ ($I_{\text{Single\_Reff}}$). Based on the study results it could be expected that the application of an objective function that addresses multiple aspects of a hydrograph improves runoff simulations for calculating a suite of SFCs. Calibration approaches based on simulating the general shape of the hydrograph ($I_{\text{Multi}}$, $I_{\text{Multi\_Reff}}$

and $R_{\text{eff}}$) reveal distinct results regarding individual SFCs, $R_{\text{eff}}$ and MARE. $R_{\text{eff}}$, and to a lesser extent MARE, are improved with the more weight $R_{\text{eff}}$ has in model calibration, whereas SFC estimates tend to be more accurate when SFCs are part of the objective function (in combination with $R_{\text{eff}}$). The results confirm that the objective functions $I_{\text{Multi}}$ and $I_{\text{Multi\_Reff}}$ constrain the model better for simulating the general shape of the hydrograph and thus are more suited for model simulations aiming at many different SFCs than SFC-specific model calibrations. However, considering that SFCs not incorporated in the

objective function showed little change in estimation error brings into question the benefit of including SFCs into model calibration instead of applying a traditional calibration approach based on $R_{\text{eff}}$. Calibrating a runoff model for estimating many different SFCs from one single hydrograph becomes a trade-off between finding a parameterization that is general enough to represent different aspects of the hydrograph and that simultaneously emphasizes specific SFCs. As stated by Caldwell et al. (2015), there is little chance to find an objective function suitable to estimate all SFCs because fitting model

parameters to some hydrograph aspects inevitably disregards other aspects. Similar conclusions were drawn by Zhang et al. (2016) who calibrated a runoff model with a multi-objective function consisting of 16 SFCs of interest to capture an overall flow regime. While applying the multi-objective function resulted in an increased performance for low-flow and high-flow magnitudes, they reported a decreased model performance for mean-flow magnitude-related SFCs. These trade-off situations are common as perfect model parameterizations are usually not possible due to a variety of uncertainty sources, such as

model structural uncertainty and input and runoff data uncertainty (Beven, 2016), In addition, various parameterizations can also have their strengths and weaknesses for different parts of the hydrograph.

A noticeable result from the current study is the distinct difference in model performance in calibration and validation when using the objective function $I_{\text{Single}}$. While almost perfect fits are achieved in calibration for all catchments and SFCs, model errors tend to be much higher in validation with a considerable spread between catchments as well as a clear difference

depending on the SFC. This observation confirms that the model is able to simulate the SFCs well, but also outlines that a good model calibration does not imply robust simulations in validation. In general, it seems that SFCs that are strongly related to physical catchment properties (e.g. rate of streamflow recession) are the most robust, followed by SFCs representing average flow condition with a moderate robustness. SFCs that are a measure of more extreme high-flow conditions are the least robust, possibly because these conditions are subject to inter-annual weather changes and are more

difficult to model due to their dynamic behaviour. A low robustness could also indicate that the model structure might be suboptimal for some catchments.

The two least robust SFCs are MH10 and TL1. MH10 simulations with $I_{\text{Single}}$ yield by far the poorest results of all objective function types with very large normalized error in both positive and negative directions. In comparison, the high estimation errors for TL1 depend on the modelling time period. The high estimation errors for TL1 in period 2 stem from years where




the minimum runoff was simulated in late winter while the observed minimum was in late fall. By visually analyzing the temperature and runoff time series, it can be hypothesized that such model simulations mainly happened in years with successive weeks of continuously little precipitation during late winter. Such prolonged drier periods occurred more often in one of the two modelling time periods and thus evoked the distinct bias in model accuracy depending on the simulation period. Both TL1 and MH10 are calculated from a single value per year, as opposed to e.g. RA7, which is based on all recessions. In model calibration, many parameter sets are derived that perfectly simulate this single value. However, a good simulation of either TL1 or MH10 is not so much dependent on an accurate representation of dominant runoff processes. Thus, model results for the validation period using input data of identical quality can fail to accurately simulate either SFC because of parameter sets 'tuned' to the data as opposed to being based on modelling the process.

## 4.2 Model performance regarding SFCs

The runoff model tends to underestimate SFCs related to mean and high-flow conditions, while SFCs representing low-flow conditions are generally overestimated. These results are consistent with those of Olsen et al. (2013), Caldwell et al. (2015), and Vis et al. (2015) and can partly be explained by the model behaviour characterized by a less pronounced runoff response to precipitation events but increased groundwater discharge to the stream during drier periods compared to the observed data (Vis et al., 2015). The observations that average flow conditions are better simulated than extremes (Caldwell et al., 2015; Vis et al., 2015) or that high-flow related SFCs are more accurately estimated than those related to low flow (Shrestha et al., 2014; Ryo et al., 2015) cannot be confirmed with our results. None of these earlier studies explicitly included SFCs into model calibration and the deviating results could be attributed to the differing approaches to defining the objective function(s). This presumption is supported by the previously described differences in results of Vis et al. (2015) although they applied the same runoff model, catchments and SFCs.

## 4.3 How to select SFCs for a multi-index calibration approach

The current study supports the assumption that including SFCs into model calibration helps to preserve most hydrograph aspects relevant to those SFCs. Thus, an objective function based on several SFCs is expected to result in a hydrograph from which a suite of SFCs can be calculated. Not knowing which SFCs will be relevant for a given study, a guideline as to which SFCs the model calibration could be based on would be helpful. The first step towards a guideline consists of selecting SFCs that are potentially valuable for model calibration. This selection was based on the concept of robustness and information value of SFCs, which is comparable to the approach used by Euser et al. (2013) who assessed the realism of model structures. Like Euser et al. (2013), results from the current study indicated that high robustness was not necessarily related to high information value, emphasizing the importance of selecting SFCs by jointly evaluating robustness and information value.

A model calibrated on certain flow conditions (low, medium and high flow) is beneficial for SFCs representing these flow conditions (see e.g. Murphy et al., 2013), so it was hypothesized that the information value of the selected SFCs is highest



for SFCs belonging to the same group of flow conditions. Surprisingly the results did not reveal any pattern related to flow conditions and thus no recommendation for the final selection of SFCs can be made. Since this study was based on a limited number of SFCs it could be interesting to test the hypothesis by analyzing a greater number of SFCs. Testing a larger number of SFCs might reveal relations that are difficult to see with a small sample. Furthermore, more knowledge about the effect of

single SFCs or the combination of SFCs used as objective functions on runoff simulations could be gained by using synthetic data and a modelling approach where an excellent hydrograph fit is possible (e.g. HBV-land in Seibert and Vis, 2012).

### 4.4 Objective functions, their estimation accuracy and consequences for practical applications

The emphasis of SFC-related modelling studies changed in recent years from estimating single SFCs to simulating a suite of SFCs (Olden and Poff, 2003). The modelling design of this study combined both approaches for the same SFCs and

catchments and thus enabled a direct comparison of the results. Ideally, the runoff model could be calibrated to simulate a hydrograph for each catchment from which any SFC can be calculated. Such an approach ensures a relatively small calibration effort, which is especially valuable if one is interested in modelling many catchments and/or various scenarios. However, results indicate that SFCs related to a more generally calibrated model (e.g. $R_{eff}$, $I_{Multi}$ or $I_{Multi\_Reff}$) are less accurate than when they are estimated from hydrographs based on targeted model calibrations (e.g. $I_{Single}$ or $I_{Single\_Reff}$). This fact has

substantial implications for the later application of simulated SFCs related to flow alteration – ecosystem change relationships. As stated by Carlisle et al. (2010), with high errors in SFC estimates, only considerable flow departures from natural conditions can be detected. Also, inaccurate SFC values can impede the generation of more robust relationships that are ultimately needed for sustainable flow management guidelines (Arthington et al., 2006; Poff and Zimmermann, 2010; Gillespie et al., 2015). As with regional statistical approaches, incorporating SFCs into model objective functions implies

that a modeller knows which SFCs are relevant and that the model must be recalibrated if one is interested in additional SFCs. The advantage of runoff models over multivariate regressions and observed streamflow series includes their use for climate scenario analysis or for simulating runoff in ungauged catchments with the latter being one of the ultimate aims in the ELOHA framework (Poff et al., 2010). Modelling SFCs gets even more challenging when moving from a gauged to an ungauged catchment. An appropriate calibration strategy targeted to the main simulation goal is crucial for any subsequent

regionalization.

### 4.5 Choice of the runoff model for estimating SFCs

When comparing SFCs estimated from simulations of different runoff models (e.g. HBV, Precipitation runoff modelling system (PRMS), etc.), the question can be raised whether the results depend on the selected model. This question is especially important for resource managers who need to make decisions based on model results from different studies

(Caldwell et al., 2015). A comparison of runoff models with different spatial scales that rely on different data inputs was conducted by Caldwell et al. (2015). Their results do not indicate that a certain runoff model is more suited for predicting





SFCs than others, but rather that the calibration process probably has as much influence as the model structure. Thus, it can be assumed that the conclusions of this study would be similar if a different calibrated runoff model was applied.

## 5    Conclusions

In this study, we evaluated the value of using SFCs for the calibration of a runoff model used to estimate SFCs. The results suggest that the choice of the objective function used for model calibration strongly influences the estimation accuracy of SFCs. While the model was capable of correctly simulating any of the tested SFCs, a good reproduction of a particular SFC was generally achieved when this SFC was included in the objective function. SFC estimates from model simulations with an objective function consisting of a representative selection of SFCs resulted in comparable accuracies to the estimates from model runs based on the commonly used Nash–Sutcliffe efficiency when evaluated against SFCs not included in the objective function. Estimates of SFCs that are less dependent on the short-term weather input or SFCs representing average flow conditions were more robust than other SFCs. Since the results imply that one has to consider significant uncertainties when simulated time series are used to derive SFCs that were not included in the calibration, we strongly recommend calibrating the runoff model explicitly for the SFCs of interest.

### Data availability

Data used in this study is available at the U.S. Department of Commerce (2007a, 2007b,) and the U.S. Geological Survey (2016a, b).

### Author contributions

Sandra Pool, Marc Vis, Rodney Knight and Jan Seibert designed this study based on a previous collaboration; Marc Vis performed the runoff simulations; Sandra Pool analyzed the results that were discussed with all coauthors. Writing of the paper was led by Sandra Pool with contribution of all coauthors.

### Competing interests

The authors declare that they have no conflict of interest.

### Acknowledgements

This paper is a product of discussions and activities that took place at the U.S. Geological Survey John Wesley Powell Center for Analysis and Synthesis as part of the workgroup focusing on Water Availability for Ungauged Rivers (https://powellcenter.usgs.gov/). Funding for this research was provided by the U.S. Geological Survey Cooperative Water



Program and the University of Zurich. Any use of trade, firm, or product names is for descriptive purposes only and does not imply endorsement by the U.S. Government.

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



**Table 1.** Description of streamflow characteristics used to calibrate the runoff model (adapted from Knight et al., 2014; U.S. Geological Survey, 2014) [mm d-1, millimeters per day; -, no units; a-1, per annum; %, percent]

| Streamflow characteristic | Abbre­viation | Definition | Flow condition | Unit |
|---|---|---|---|---|
| *Magnitude* | | | | |
| Mean annual runoff | MA41 | Annual mean daily streamflow | mean-flow | [mm d$^{-1}$] |
| Maximum October runoff | MH10 | Mean maximum October streamflow across the period of record | high-flow | [mm d$^{-1}$] |
| Lowest 15% of daily runoff | E85 | 85% exceedance of daily mean streamflow for the period of record | low-flow | [mm d$^{-1}$] |
| Rate of streamflow recession | RA7 | Median change in log of streamflow for days in which the change is negative across the period of record | mean-flow | [mm d$^{-1}$] |
| *Ratio* | | | | |
| Average 30-day maximum runoff | DH13 | Mean annual maximum of a 30-day moving average streamflow divided by the median for the entire record | high-flow | [-] |
| Base flow | ML20 | Ratio of total base flow to total flow. Base flow is the minimum flow in a 5-day window if 90% of that flow is less than the minimum of the 5 day-window before and after the considered block | low-flow | [-] |
| Stability of runof | TA1 | Measure of the constancy of a flow regime by dividing daily flows into predetermined flow classes | mean-flow | [-] |
| *Frequency* | | | | |
| Frequency of moderate floods | FH6 | Average number of high-flow events per year that are equal to or greater than three times the median annual flow for the period of record | high-flow | [a$^{-1}$] |
| Frequency of moderate floods | FH7 | Average number of high-flow events per year that are equal to or greater than seven times the median annual flow for the period of record | high-flow | [a$^{-1}$] |
| *Variability* | | | | |
| Variability of March runoff | MA26 | Standard deviation for March streamflow over the period of record divided by the mean streamflow for March over the period of record | mean-flow | [%] |
| Variability in high-flow pulse duration | DH16 | Standard deviation for the yearly average high-flow pulse duration (daily flow greater than the 75$^{th}$ percentile) divided by the mean of the yearly average high-flow pulse duration multiplied by 100 | high-flow | [%] |
| Variability of low-flow pulse count | FL2 | Standard deviation for the average number of yearly low-flow pulses (daily flow less than the 25$^{th}$ percentile) divided by the mean low-flow pulse counts multiplied by 100 | low-flow | [%] |
| *Date* | | | | |
| Timing of annual minimum runoff | TL1 | Julian date of annual minimum flow occurrence | low-flow | [Julian day] |





**Table 2.** Objective functions used in model calibration. Objective functions were calculated with observed (obs) and simulated (sim) runoff ($Q$) or SFCs ($I$).

| Objective function | Abbreviation | Definition | Optimal value |
|---|---|---|---|
| Nash-Sutcliffe efficiency | $R_{\text{eff}}$ | $1 - \dfrac{\sum(Q_{\text{obs}} - Q_{\text{sim}})^2}{\sum(Q_{\text{obs}} - \overline{Q_{\text{obs}}})^2}$ | 1 |
| Efficiency for each individual SFC[1] | $I_{\text{Single}}$ | $1 - \dfrac{|I_{\text{obs}} - I_{\text{sim}}|}{I_{\text{obs}}}$ | 1 |
| SFC and Nash-Sutcliffe efficiency | $I_{\text{Single\_Reff}}$ | $0.5\,(I_{\text{Single}} + R_{\text{eff}})$ | 1 |
| Efficiency for the selected SFCs[2] | $I_{\text{Multi}}$ | $0.25\,(I_{\text{Single}_1} + \ldots + I_{\text{Single\_n}})$ | 1 |
| SFCs and Nash-Sutcliffe efficiency | $I_{\text{Multi\_Reff}}$ | $0.8\,I_{\text{Multi}} + 0.2\,R_{\text{eff}}$ | 1 |

[1]For each of the 13 SFCs a specific $I_{\text{Single}}$ exists.

[2]$I_{\text{Multi}}$ consists of the $n$ most robust and informative SFCs.



**Table 3.** Performance measures used in model evaluation. Performance measures were calculated with observed (obs) and simulated (sim)
5  runoff ($Q$) or SFCs ($I$).

| Performance measure | Abbreviation | Definition | Optimal value |
|---|---|---|---|
| Nash-Sutcliffe | $R_{\mathrm{eff}}$ | $1 - \dfrac{\sum(Q_{\mathrm{obs}} - Q_{\mathrm{sim}})^2}{\sum(Q_{\mathrm{obs}} - \overline{Q_{\mathrm{obs}}})^2}$ | 1 |
| Mean absolute relative error[1] | MARE | $1 - \dfrac{1}{n}\sum \dfrac{\lvert Q_{\mathrm{obs}} - Q_{\mathrm{sim}}\rvert}{Q_{\mathrm{obs}}}$ | 1 |
| Normalized SFC error[2] | nSFC | $\dfrac{I_{\mathrm{obs}} - I_{\mathrm{sim}}}{R_{\mathrm{obs}}}$ | 0 |

[1] $n$ is the number of days.

[2] $R$ is the range of possible values of a SFC for the respective catchment.





**Figure 1.** Location of the 25 study catchments in the Tennessee River basin (Table 1 in Vis et al. (2015) for more information).




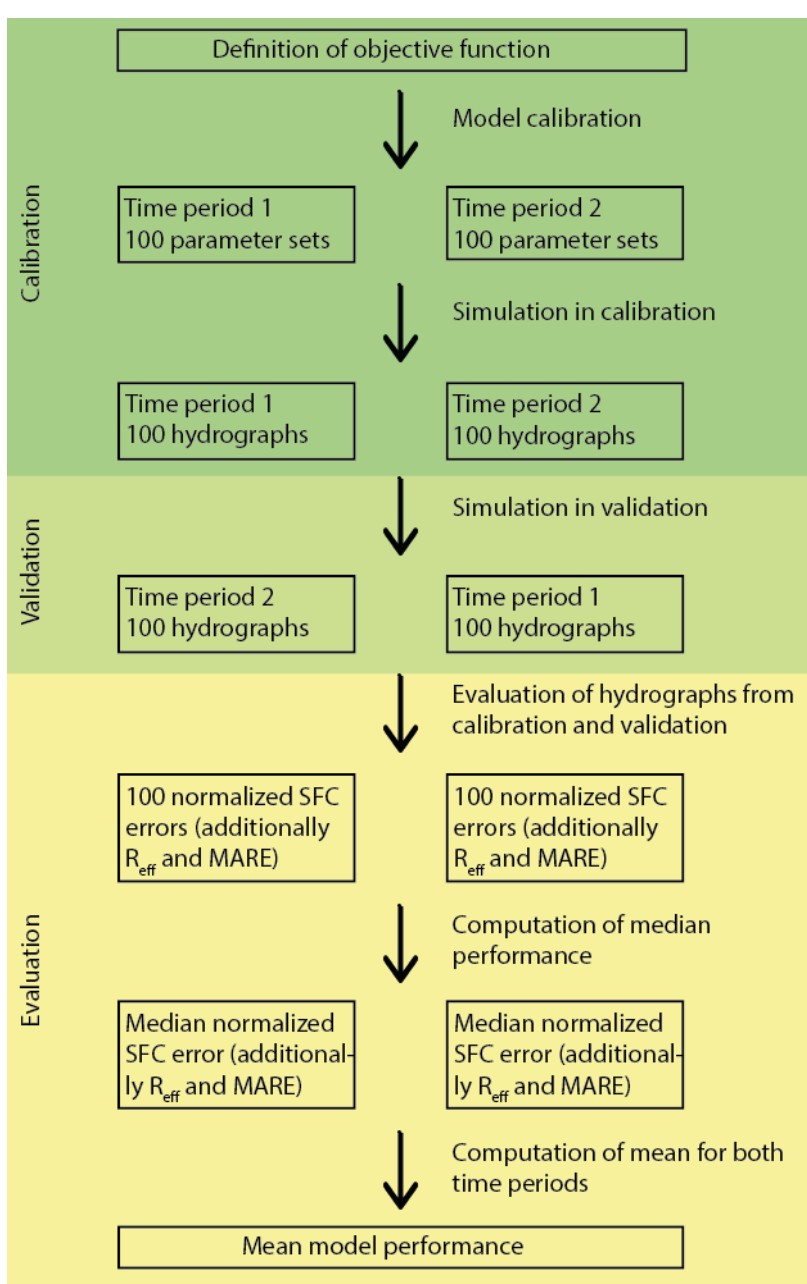

**Figure 2.** Flow chart of the modelling approach consisting of calibration, validation and evaluation in time period 1 (1984 - 1996) and time period 2 (1997 - 2009) and completed for each of the five objective function types $R_{eff}$, $I_{Sinlge}$, $I_{Single\_Reff}$, $I_{Multi}$, $I_{Multi\_Reff}$.





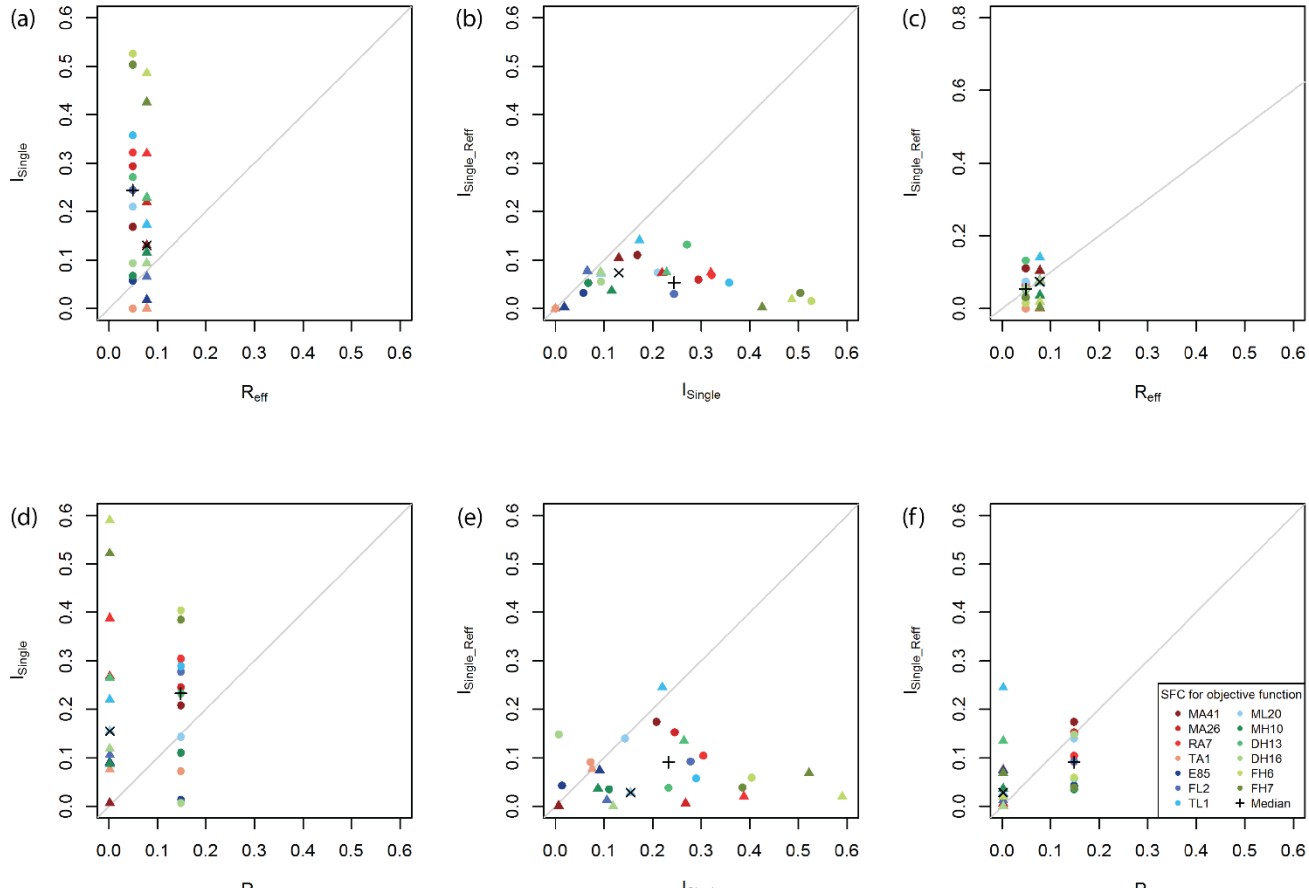

**Figure 3.** Comparison of absolute normalized TA1 error (nSFC) in calibration (a-c) and validation (d-f) calculated from model calibrations with the objective functions $R_{eff}$, $I_{Single}$ and $I_{Single\_Reff}$. Absolute normalized SFC errors correspond to the median of the 25 catchments and are shown separately for both modelling time periods (triangles for period 1 (1984 - 1996) and circles for period 2 (1997 - 2009)). The x and plus symbols represent the median of period 1 and period 2 respectively.




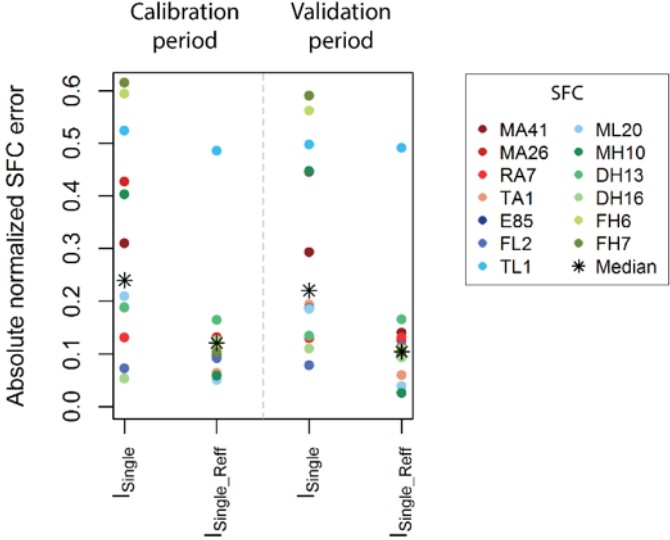

**Figure 4.** Absolute normalized SFC error (nSFC) for the model calibration (left side) and model validation periods (right side) calculated from model calibrations with the objective functions $I_{Single}$ and $I_{Single\_Reff}$. Values correspond to the median error of all 13 objective function versions and were calculated from the median of the 25 catchments and the mean of both modelling time periods.





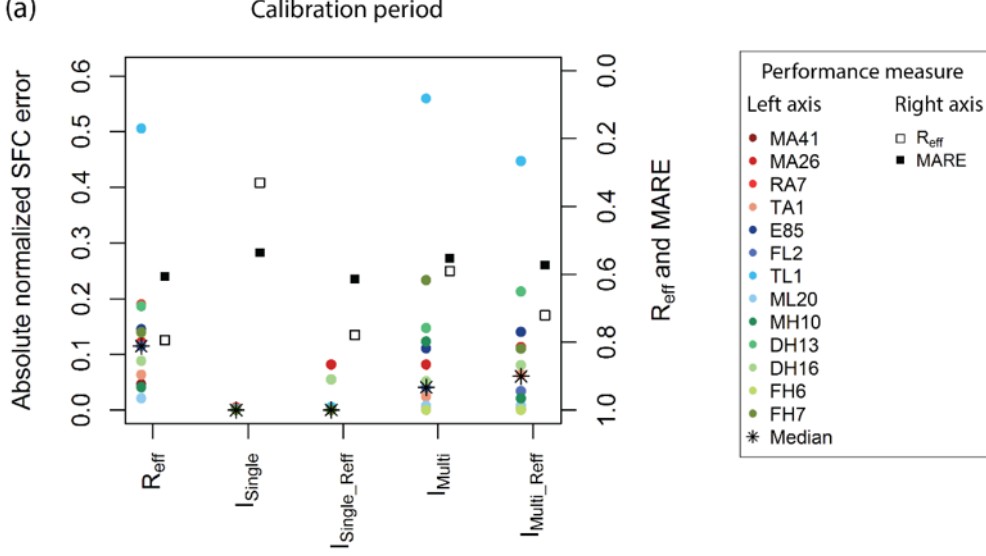

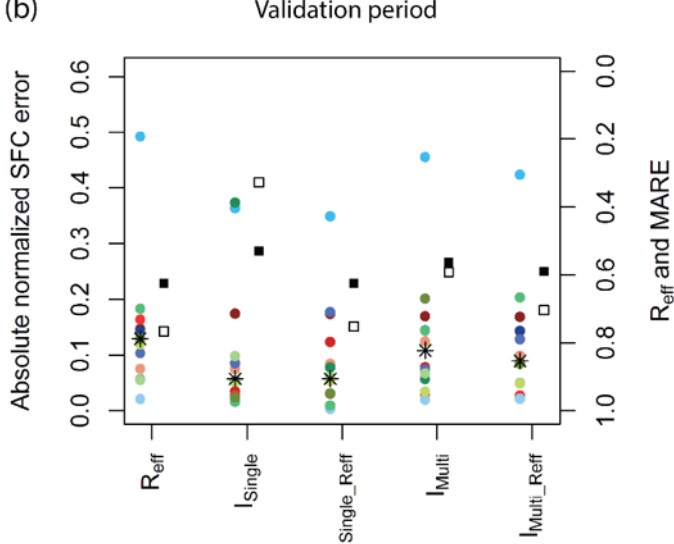

**Figure 5.** Model performance in a) calibration and b) validation for absolute normalized SFC errors (nSFC) as well as $R_{eff}$ and MARE depending on the the objective function used in calibration (optimal value is one for $R_{eff}$ and MARE and zero for all SFC related performance measures). Model performance values correspond to the median of the 25 catchments and the mean of both modelling time periods. $R_{eff}$ and MARE values for the objective functions $I_{Single}$ and $I_{Single\_Reff}$ were calculated as the median over all 13 versions. Note that in calibration with $I_{Single}$ and $I_{Single\_Reff}$ the values of all or most absolute normalized SFCs plot at the same value.





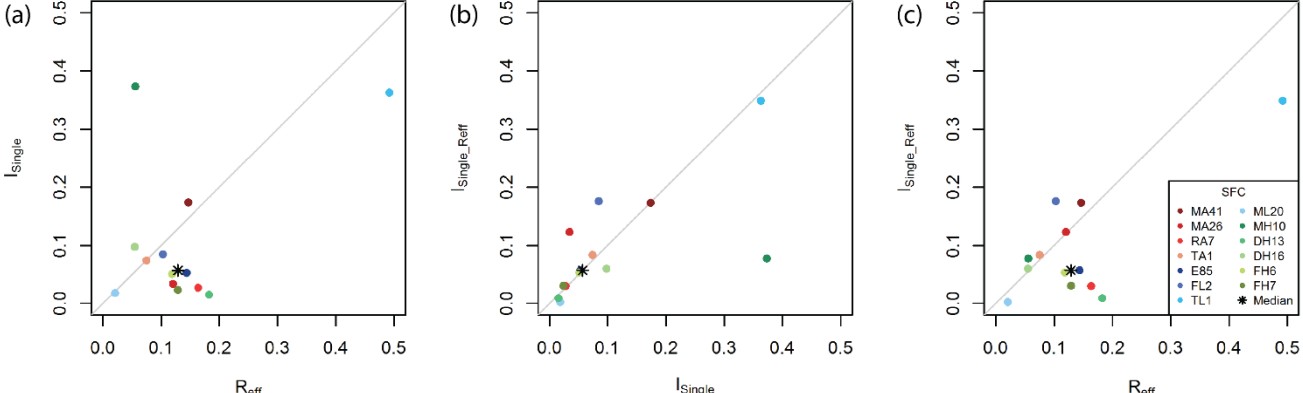

**Figure 6.** Comparison of absolute normalized SFC errors (nSFC) in validation calculated from model calibrations with the objective functions $R_{eff}$, $I_{Single}$ and $I_{Single\_Reff}$. Absolute normalized SFC errors correspond to the median of the 25 catchments and the mean of both modelling time periods.

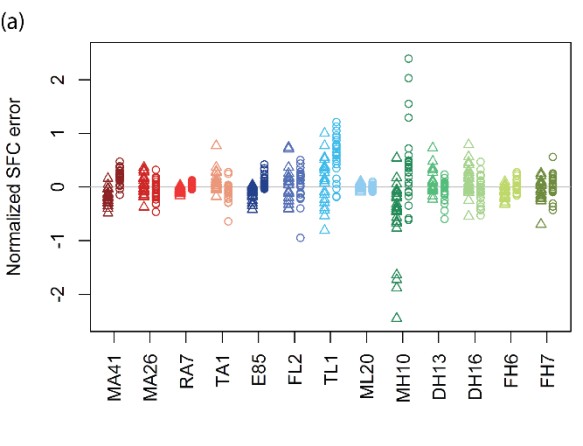

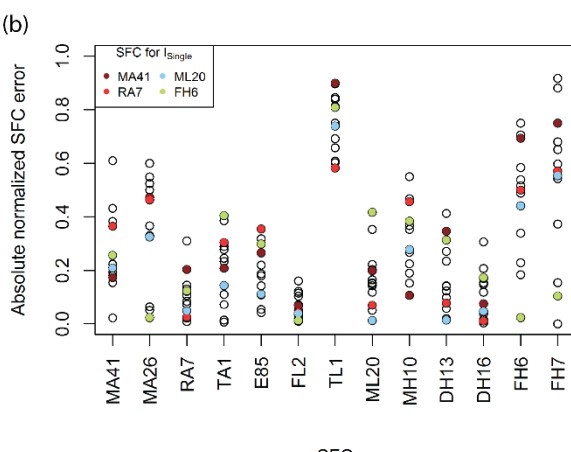

**Figure 7.** a) Robustness: normalized SFC errors (nSFC) in validation calculated from model calibrations with the objective function $I_{Single}$
10  for the respective SFC. Values are shown for all 25 catchments and both modelling time periods (triangles for period 1 (1984 - 1996) and circles for period 2 (1997 - 2009)). b) Information value: absolute normalized SFC errors (nSFC) in validation calculated from model calibrations with all 13 objective functions $I_{Single}$. Model performance values correspond to the median of the 25 catchments and the mean of both modelling time periods. Each circle represents a SFC used for $I_{Single}$. The coloured circles show the information value of the final selection of SFCs for the objective function $I_{Multi}$.




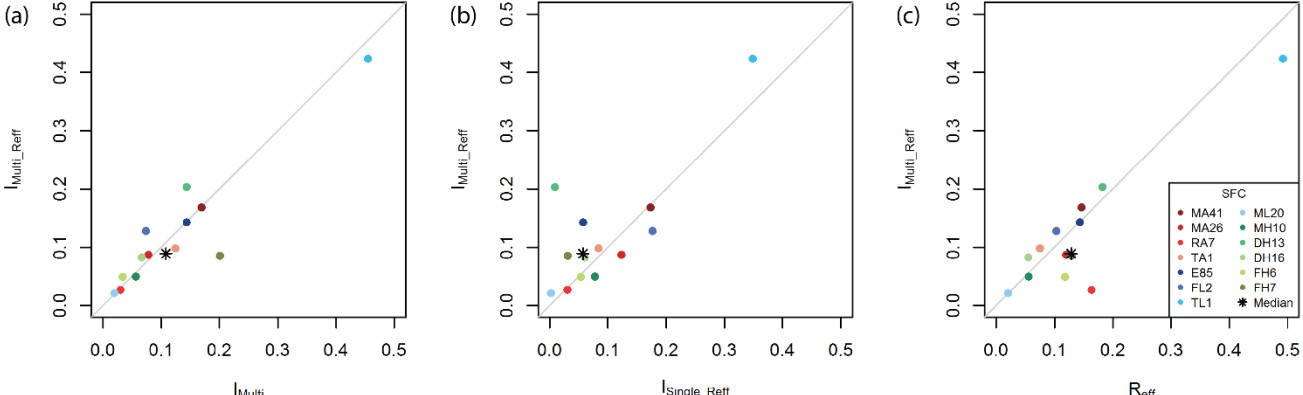

**Figure 8.** Comparison of absolute normalized SFC errors (nSFC) in validation calculated from model calibrations with the objective functions $R_{eff}$, $I_{Single\_Reff}$, $I_{Multi}$ and $I_{Multi\_Reff}$. Absolute normalized SFC errors correspond to the median of the 25 catchments and the mean of both modelling time periods.

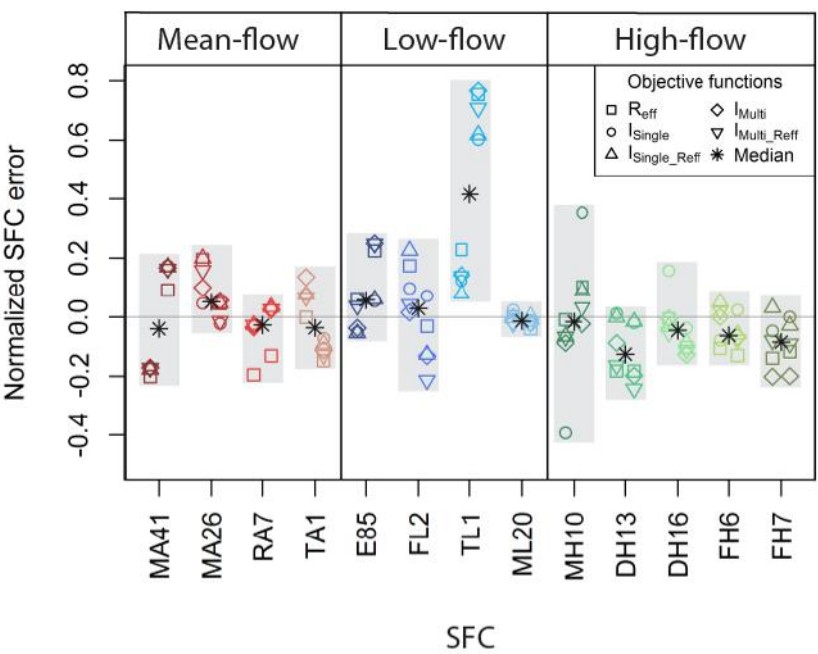

**Figure 9.** Normalized SFC errors (nSFC) in validation depending on the objective function used in calibration. Model performance values correspond to the median of the 25 catchments and are shown for both modelling time periods (period 1 (1984 - 1996) on the left side and period 2 (1997 - 2009) on the right side).





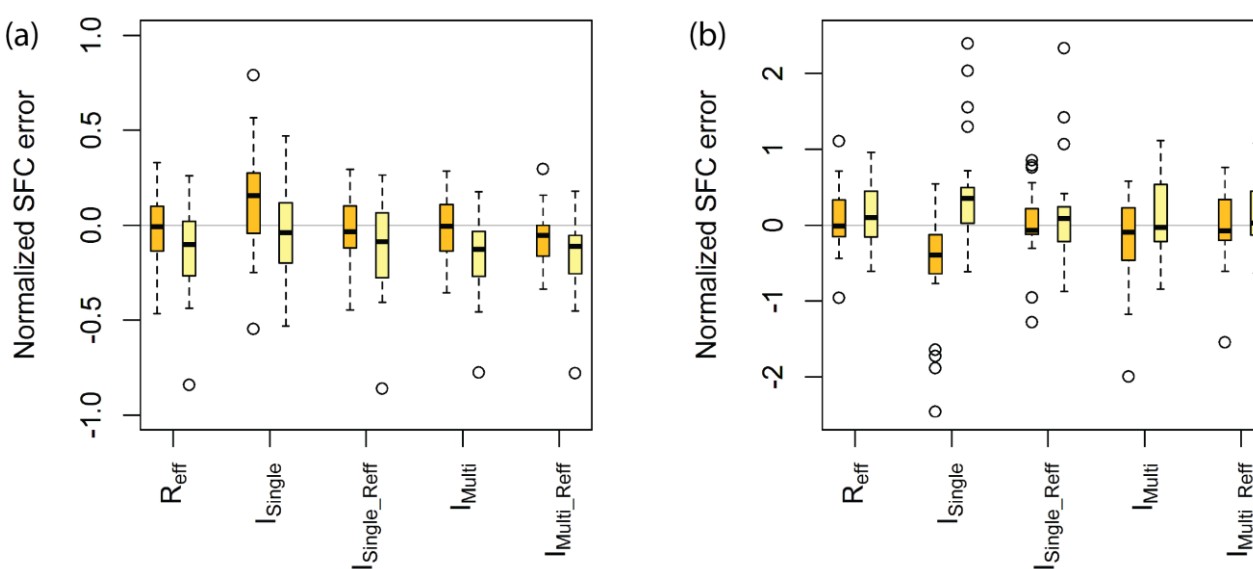

**Figure 10.** a) Normalized DH16 errors (nSFC) and b) normalized MH10 errors (nSFC) in validation depending on the objective function used in calibration. Absolute normalized SFC errors are shown for all 25 catchments and for both modelling time periods (period 1 (1984 - 1996) in orange on the left side and period 2 (1997 - 2009) in yellow on the right side). Note the difference in y-axis.