# Peer review of "Streamflow characteristics from modelled runoff time series importance of calibration criteria selection"

_Hydrology and Earth System Sciences, 2016_

## Referee Comment (RC1) · Anonymous Referee #1 · 25 Nov 2016

It is a very good idea and a well know idea, that the models should be calibrated using criterions relevant to the purpose of the model and simulations. The criterions here are many but seems relevant to ecological studies. The relevance of the choosen SFC should be argued for in relevance to the purpose. Here are maybe to many SFC and the it becomes difficult to keep track of them. Are they correlated, are they in contradiction (to fulfill one means that another suffer) etc. It might be an idea to pick the most important for the kind of studies the model should be used in and discuss this closer.

But before it is relevant to discuss other criterions and SFC used for calibration and how well these can be recreated you need to demontrate that teh model is able to reproduce

observed flow to a certain degree. I can not see that this is the case here. As long as this is not the case and is demontrated the remaining work becomes irrelevant. If the model is not able to get an Reff higher than 0.7 the interesting discussion is then if it is able to or relevant to use for simulation of other SFC.

It might be that you are able to acheive good simulations in this catchment. I that case show this also by showing hydrographs. If you are not able to achieve a Reff up to 0.7 you need to disuss why first and then move to other STC that you can argue for the model is able to recreate in spite of a poor Reff. If the Reff is OK, but as you initially says is not enough for some studies, you can compare calibration with other criterions againt Reff and each other. I beileve that it is the latter that you try to do, but as long as I as a reader am not able to see that your model and data actually are representative or good enough to reporduce the observed the discussion becomes not relevant or interesting.

So before revising this deeper several clification have to be made initially. But the topic is very intersting and relevant so I hope you are able to structure this in a way that make many readers interested and enlightened.

Please also note the supplement to this comment:
http://www.hydrol-earth-syst-sci-discuss.net/hess-2016-546/hess-2016-546-RC1-supplement.zip

---

## Short Comment (SC1) · 25 Nov 2016

Thanks for these comments. Here just a quick clarification, before we will respond in more detail: The HBV model is capable of reproducing the observed runoff for these catchments reasonably well when calibrated on the efficiency (Reff). As stated in the preceding paper (Vis et al., 2015) using the same data set:

"The model efficiencies that could be achieved for the different catchments varied from 0.64 to 0.91 (calibration) and 0.61 to 0.90 (validation), indicating reasonably good runoff simulation with the calibrated HBV-light model."

This means that it is in principle possible to get good performances in terms of Reff or

in other words, poorer performances are a result of the respective calibration criteria rather than an inappropriate model (structure).

Best regards, Jan Seibert

Vis, M., R. Knight, S. Pool, W. Wolfe, J. Seibert, 2015. Model calibration criteria for estimating ecological flow characteristics, Water, 7(5), 2358-2381; doi:10.3390/w7052358.

---

## Referee Comment (RC2) · Anonymous Referee #1 · 28 Nov 2016

This is good news. When I read the paper it was not obvious to me that this was the case. This probably seem clear to the writer, but to a reader not knowing about the previous work with the same model this was not so clear.

From my understanding the basic idea behind this paper is that for many purposes the commonly used Reff is not a good enough calibration criteriona and you suggest different new approaches that can be used seperately or combined with Reff.

You mention in generalm terms that high peaks and low flow is not well enough simulated. Could this and other features the "good Reff model" does not capture be illustrated in the introduction as a background for the study and choice of additional criterions?

The paper would also benefit of a short argumentation for the choice of SFC as calibration criterions. And also how these are calculated. Ieff does not tell med how you find the value for the criterions and the table does not say so either.

There are other comments in the paper that I hope make sense to you. And I look forward to see a revised version. I assume these suggested changes also will influence the discussion and that we will have another round on this when ready.

---

## Referee Comment (RC3) · B. Guse (Referee) · 1 Dec 2016

Review of the manuscript "Streamflow characteristics from modelled runoff time series – importance of calibration selection criteria" by Pool et al.

In this article, Pool et al. investigated the benefit of using streamflow characteristics (SFC) for model calibration and showed how the model calibration can be improved in comparison to the use of the Nash-Sutcliffe-Efficiency (Reff). They proposed a joint approach of using several SFCs and the Reff. Hereby the selection of the specific SFC depends on the goal of the study.

I like the idea of the article and see its potential after revision.

[Figure]

Major comments:

In its current state, the article is in my opinion mainly related to hydrology (also by the selection of the journal). At several parts the authors emphasized its ecological importance (e.g. Page 1, Line 10, P.2, L.1-2, P.11, L.14-16). However, I do not really see a connection to ecology. Thus, either the article has to be fully focused on hydrology or it is required to emphasize its ecological relevance

In this article, the Nash-Sutcliffe Efficiency is used as an example for a traditional calibration approach. However, in recent studies the use of several performance measures related to different parts of the hydrological system is recommended. In particular the use of typical performance metrics such as Reff or Kling-Gupta-Efficiency (KGE) in combination with signature measures is recommened (see eg. Van Werkhoven et al., 2009). Thus, I think that a comparison only with the Reff is not sufficient. I recommend to use two or three performance measures such as PBIAS or KGE to show that SFCs also outperforms a calibration approach based on NSE in combination with PBIAS (or other performance measures).

Moreover, it needs at least to be discussed how a calibration approach based on these SFCs is related to recent studies using hydrological signatures such as segments of the FDC (see Yilmaz et al., 2008, Pfannerstill et al., 2014).

P. 1, L. 24-27: I do not fully agree with this statement. There are different ways of how to calibrate a discharge time series. Certainly there are studies which are focused on certain parts such as on high or low flows. However, other studies aim to represent the whole hydrological system at best without neglecting or emphasizing certain parts. In the way towards a good representation of the hydrological system, the latter one should be the general goal and a strong focus on certain parts of the hydrological system should be a specific case.

P.2, L. 21: Here, SFCs are defined "as equivalent to hydrological signatures". In this case, I do not understand the use of SFC. It is stated before that hydrological signatures

are a more common term in hydrology. It is certainly required to justify why you used the term SFC since I do not really see the strong relationship to ecology.

I think that the article would benefit from a more detailed interpretation of the results. For example on P. 7. L. 12-15; P.8, L.17-18: Can you explain these results or more specifically the behaviour of these SFCs?

It could be interesting to analyze the relationship/correlation between the SFCs.

The combination of different metrics might outperform in general single-metric approaches. However, the more metrics are included, the more a trade-off might occur and the equifinality problem arises. In this context, can you give a recommendation for a good number of required SFCs? In the best case, a systematic way of how to select the best SFCs will be provided. Even though when I expect that it is difficult to find a precise number, it is worth discussing this point.

The figures 3-8 are very similar (at least visually). I think that the article would benefit from emphasizing the relevance of each figure. It is partly difficult to differentiate them. Maybe you can also thinking about reducing the number of figures to improve the overall message. For example, the figures 6 and 8 have almost the same figure caption. To summarize this point, it is easier to detect the whole message in the case of a more distinct presentation of the results. One example for this is the figure 9 which can be clearly distinguished from the other figures. These results are easier to understand.

P. 8, L.12-13: Could you specify how you can here differentiate between error dependence on time period or objective function?

P.8, L.14-15: I do not understand this statement that the SFCs are neither related to flow components nor to flow conditions. Hydrological signatures (as an equivalent term) are known to be of special importance to explain the hydrological behaviour. Thus, what can we learn from using these SFCs in terms of the hydrological behaviour in the catchment. And how is this related to the general idea of the hydrological signatures?

P.8, L.26 to P. 9, L.21: I agree with this part which is clearly understandable, but certainly also not surprising. It is mostly existing knowledge of hydrological modelling. What can we learn here except of using several and different metrics. I recommend to shorten this part and emphasize the most important points from this study. In contrast, I really like to following passage (P. 9, L.21-31).

Please also discuss the impact of a SFC-based calibration for the process representation. Can you state that the hydrological system is overall better represented by using several SFCs?

Please discuss the benefit of optimizing one specific SFC. This leads to a modelled hydrograph which is able to represent a very specific condition but probably not the overall hydrograph. This implies that the part of the hydrological system which is not in the focus of this SFC is probably not adequately considered. This might be of particular relevance when using very specific SFCs such as MA26.

Minor comments:

P.1, L. 12: maybe "optimization" instead of "minimization or maximization".

P.1., L.16: Are these over- and underestimations a general aspect of these SFCs or are they case-specific?

P.2, L.16-19 and L. 30: I recommend to include the study from Yilmaz et al. (2008).

P.2, L. 34: The meaning of "esoteric and subtle aspects of the flow regime" is unclear.

P.3, L. 19: Please think about renaming the section to "Methods and materials", since a catchment is not a method.

P.5, L.31: Why you have used 0.2 and 0.25 as weights?

P.6, L. 9: Could you specify "median parameter set"?

P. 11, L.15: Could you specify "later application of simulated SFCs related to flow alteration – ecosystem change relationships". This aspect was up to now neither in the focus of the article nor emphasized as an overall aim.

Table 1: TA1: runoff with two f

Table 1: Why you have named the SFCs FH6 and FH7 and not FH3 and FH7.

Fig. 5: Can you explain the outlier TL1?

References:

Pfannerstill, M.; Guse, B. and Fohrer, N. (2014): Smart low flow signature metrics for an improved overall performance evaluation of hydrological models. J. Hydrol. 510: 447–458.

van Werkhoven, K., Wagener, T., Reed, P., Tang, Y. (2009): Sensitivity-guided reduction of parametric dimensionality for multi-objective calibration of watershed models. Adv. Water Resour. 32 (8), 1154–1169.

Yilmaz, K. K., Gupta, H. V., and Wagener, T. (2008): A process-based diagnostic approach to model evaluation: Application to the NWS distributed hydrologic model, Water Resour. Res., 44, W09417, doi:10.1029/2007WR006716.

---

## Author Comment (AC2) · 5 Dec 2016

Dear reviewer,

Thank you for your efforts with our manuscript. We greatly appreciated the comments. Please see below our detailed response to each of the comments.

Best regards,

Sandra Pool and Co-Authors

**General comments RC 1**

**Comment 1:** It is a very good idea and a well know idea, that the models should be calibrated using criterions relevant to the purpose of the model and simulations. The criterions here are many but seems relevant to ecological studies. The relevance of the choosen SFC should be argued for in relevance to the purpose. Here are maybe to many SFC and the it becomes difficult to keep track of them. Are they correlated, are they in contradiction (to fulfill one means that another suffer) etc. It might be an idea to pick the most important for the kind of studies the model should be used in and discuss this closer.

**Reply 1:** We agree that we used many SFCs. There are more than 150 different SFCs used in ecohydrological studies and many of them are correlated or redundant (Olden and Poff, 2003). In our study we use 13 SFCs that have been shown to be the most relevant ones for the ecological integrity of the study catchments in previous studies (see chapter 2.2 for the motivation of the selected SFCs). We are aware that some of the selected 13 SFCs are correlated, e.g. FH6 and FH7 (we will add a correlation analysis in the revised manuscript), but still decided to use these ecologically most relevant SFCs in our study. As you suggest, selecting the most important SFC would allow a deeper analysis but also reduce the potential for general conclusions. Our 13 ecologically relevant SFCs have the advantage of representing different flow components and flow conditions.

Calibrating the model for one SFC can indeed have a negative effect on the estimation accuracy of another SFC (e.g. calibrating for FH6 (high flow) could negatively affects ML20 (low flow)) (see also our discussion in chapter 4.1). This effect seems to be inevitable in runoff modelling because model calibration is a trade-off where usually no perfect parameterization can be found due to different uncertainty sources. In case of a perfect situation with no model, parameter and data uncertainty, all SFCs could be perfectly estimated with the same runoff simulation.

Olden, J. D., Poff, N. L.: Redundancy and the choice of hydrologic indices for characterizing streamflow regimes, River Research and Applications, 19(2), 101-121, doi:

10.1002/rra.700, 2003.

**Comment 2:** But before it is relevant to discuss other criterions and SFC used for calibration and how well these can be recreated you need to demontrate that teh model is able to reproduce observed flow to a certain degree. I can not see that this is the case here. As long as this is not the case and is demontrated the remaining work becomes irrelevant. If the model is not able to get an Reff higher than 0.7 the interesting discussion is then if it is able to or relevant to use for simulation of other SFC.

**Reply 2:** Thank you for this useful comment. We agree that it is important to know that HBV can reproduce observed runoff to an acceptable degree. Based on your feedback we realized that we didn't clearly formulate this, and especially the presentation and labels of some of the figures were misleading. We will adapt the text and figures where needed to avoid any possible misunderstanding by the reader.

The HBV model is capable of reproducing the observed runoff for the study catchments reasonably well when calibrated on the Nash-Sutcliffe efficiency (Reff). Figure 5a and 5b show that the median Nash-Sutcliffe efficiency of the 25 catchments is 0.79 in calibration and 0.77 in validation when calibrated with Nash-Sutcliffe (note that values corresponding to Reff are on the y-axis at the right side of the plot). Over all 25 catchments the Nash-Sutcliffe efficiency ranges from 0.68-0.89 in calibration (24 of 25 catchments with efficiencies larger than 0.7) and 0.62-0.86 in validation (21 of 25 catchments with efficiencies larger than 0.7).

This means that it is in principle possible to get good performances in terms of Reff, or, in other words, poorer performances are a result of the respective calibration criteria rather than an inappropriate model (structure).

**Comment 3:** It might be that you are able to acheive good simulations in this catchment. I that case show this also by showing hydrographs. If you are not able to achieve a Reff up to 0.7 you need to disuss why first and then move to other STC that you can argue for the model is able to recreate in spite of a poor Reff. If the Reff is OK, but

as you initially says is not enough for some studies, you can compare calibration with other criterions againt Reff and each other. I beileve that it is the latter that you try to do, but as long as I as a reader am not able to see that your model and data actually are representative or good enough to reporduce the observed the discussion becomes not relevant or interesting.

**Reply 3:** Yes, we indeed calibrated the model with different objective functions and then compared the resulting model efficiencies.

As discussed in comment 2 of RC1, HBV was able to reasonably well reproduce the hydrographs of the 25 catchments. We decided to only report the Nash-Sutcliffe efficiencies and not to show the hydrographs. The hydrographs of the complete simulation period from 1984-2009 do not provide much information when shown in one graph (low visibility of information) and the details of one year might not be representative. However, the Nash-Sutcliffe efficiency is an integrated measure of the hydrograph fit over all years.

**Comment 4:** So before revising this deeper several clification have to be made initially. But the topic is very intersting and relevant so I hope you are able to structure this in a way that make many readers interested and enlightened.

**Reply 4:** We address your more detailed comments in the parts below (see detailed comments RC 1)

**General comments RC 2**

**Comment 1:** You mention in generalm terms that high peaks and low flow is not well enough simulated. Could this and other features the "good Reff model" does not capture be illustrated in the introduction as a background for the study and choice of additional criterions?

**Reply 1:** This is a good idea. We will extend the introduction by a part about the drawback of some commonly used calibration criteria for low and high flow related SFCs. In our previous study model calibrations with widely used objective functions such as Nash-Sutcliffe efficiency, volume error, MARE, etc. resulted in the underestimation of high-flow related SFCs by 13-33

**Comment 2:** The paper would also benefit of a short argumentation for the choice of SFC as calibration criterions. And also how these are calculated. Ieff does not tell med how you find the value for the criterions and the table does not say so either.

**Reply 2:** Table 1 in the manuscript indeed only gives a description of the SFCs used in this study. For the calculation of the SFCs the EflowStats R-Package from the USGS (chapter 2.4.3) was used. More details about the exact calculation can be found in the R scripts and the corresponding documentation. The R-package is freely available for everybody and we recommend the interested reader to check the R-package for detailed information on the calculation of the SFCs.

The 13 SFCs used in our study are commonly used SFCs of many ecohydrological studies. We discuss the choice of the SFCs in comment 1 in RC 1.

**Detailed comments RC 1: tables and figures**

**P17 L1:** Table 1- But how are these used. For instance how do you evaluate simulated against observed for TA1? This should be explained for all SFC's.

**Reply:** We use SFCs for model calibration and validation, whereby simulated SFC values are always evaluated against observed SFC value. The exact definition of the objective function as used for calibration can be found in table 2 (ISingle), whereas the performance measure used for validation is given in table 3 (nSFC). The same equations are used for comparison of simulated and observed values for all SFCs.

**P17 L3-30:** Table 1 - smaller comments on the description of SFCs

**Reply:** Thanks for these smaller comments on the description of the SFCs. We will

adapt the text in the table according to your input.

**P19 L4:** Table 3 - It is not clear for me why you need two tables for evaluation/calibration criterions. I assume you use them all in different calibrations....and then also the same in evaluating the performance? From fig 2 I get the idea. But this should be better described in the text. Reff is used both in calibration and evaluation. Describe why.

**Reply:** We decided to have two tables to make clear which criteria are used for model calibration and which criteria are used for model evaluation. We also used a separate chapter for the description of the calibration and evaluation criteria (chapter2.4.2 and 2.4.3) to make the difference more clear. Nash-Sutcliffe efficiency is the only criteria used in both calibration and evaluation. SFCs are used for model calibration and evaluation, but the exact definition is slightly different. For model evaluation the SFCs had to be normalized to make the results comparable (see chapter 2.4.3). The combined objective functions ISingle_Reff, IMulti, IMulti_Reff are only used in model calibration, whereas MARE is only used in model evaluation. We think it is helpful for the reader to keep those two tables separated, especially because the use of the SFCs is different.

As you mentioned, we use the Nash-Sutcliffe efficiency in model calibration and evaluation. The reason for that is that the Nash-Sutcliffe efficiency is an established measure used in many modelling studies in the same we as we do. It shows how well the model is reproducing the criteria it was calibrated on in an independent time period. It is also a measure of how well the general shape of the hydrograph is simulated, although with a focus on peaks. The use of Nash-Sutcliffe in our model evaluation is useful for the comparison to other studies.

**P22 L4:** Figure 3 - You refer too this figure as an illustration of variability of model perfomance...and her say it is comaprison between calibration and validation period. This is no consistent. You also say it is about A1 but shows all SFC. I do not get it. Neither that you have stablke and very poor Reff...

**Reply:** We apologize for the confusion this figure evoked. We admit that the axis labels

are not well selected and that they are misleading. We will change the axis labels to make this plot better readable. For example the axis label "Reff" will be changed to "Normalized TA1 error for models calibrated based on Reff".

For clarification we shortly describe the figure here: The figure shows the normalized TA1 error when calibrated with Reff, ISingle and ISingle_Reff.

When calibrating the model based on the Nash-Sutcliffe efficiency, we get one normalized TA1 error for each modelling time period. This results in two values of the normalized TA1 error displayed at the Reff -axis. Thus, the value of approximately 0.1 at the Reff -axis is the normalized TA1 error and not a Nash-Sutcliffe efficiency.

Since we have 13 different SFCs that we use for objective functions, we get 13 normalized TA1 error values from calibrations with ISinlge or ISingle_Reff. These are the 13 values displayed at the ISinlge-axis or the ISingle_Reff-axis. Each of the 13 different objective functions are colored based on the SFCs it is based on.

We hope that with the change of the axis labels it also becomes clear that we do not compare calibration and validation. The figure caption says that figure 5a-c are calibration results and figure 5d-f are validation results for both modelling time periods.

**Detailed comments RC 1: text**

**P1 L10:** What is the purpose? It does not come clearly forward here. Is it to simulate streamflow characteristic in ungauged basin. In that case is the simulation results relevant for ungauged basins? Is the test sites ungauged, is it split samle testing? Or is it ordinary calibrated models for guaged basin. In cas of latter, how representativ is this test then for ungauged basins?

**P1 L14:** I assume it's here it is satde what the actual purpose is. But could the strategy and the purpose be stated clearer. To a person only reading the abstract to consider if this is interesting I do not think this tells enough.

**P1 L19:** For ungauged basisn estimates or in general?

**Reply:** The three comments above all address the abstract, so we answer them in one paragraph:

We agree that the abstract should be improved to clearly state the purpose and the method of this study. We will rewrite it. For some clarification right now: In our study we only work with gauged catchments. The aim is to test whether the consideration of SFCs in model calibration improves the estimation accuracy of SFCs compared to more traditional calibration approaches using e.g. the Nash-Sutcliffe efficiency. Our results help to improve model calibration for estimating SFCs which is of great importance for a subsequent regionalization of SFCs for ungauged catchments. The ultimate aim is therefore to have improved model calibration approaches for the regionalization of runoff to ungauged catchments. For gauged catchments we don't need to model runoff for the estimation of SFCs because they can be calculated form the observed data.

**P1 L27:** I can not see many other applications for runoff simulations than recreating streamflow characteristics. so in that sense this sentence does not make sense. But if it is ecological SFC then I can see this is referred to as spesific SFC. So it migth be that ecological shoudl be added. Here and other places in the document.

**Reply:** We will change this sentence.

**P3 L9:** But initially you stated that this is what is the challange...."Ecologically relevant streamflow characteristics (SFCs) of ungauged catchments are often estimated from simulated runoff of hydrologic models." ... and the rest of the paper is about gauged basin where this challange is substantially lower... this confuses the reader as this is two very different challanges. You can discuss this but make clear already in abstract that this paper is about gauged catchments.

**Reply:** True, we will make sure in the abstract that our study is about gauged catchments (see comment P1 L10 in detailed comments RC 1 about text).

**P3 L16:** Other than?

**Reply:** We will replace ". . . or more other SFCs?" by "... or multiple SFCs?"

**P3 L17:** Spesific SFC is included also in traditional calibration (max and mean is spesific SFC). But maybe not those intersting for a spesific purpose. Understand the point, but is not clearly formulated.

**Reply:** We will replace ". . .and those where specific SFCs are included?" by ". . . and those where the SFCs of interest are included?"

**P4 L1:** Is usually a challenge if water drain out uncontrolled. Is this a problem in this catchment?

**Reply:** We agree that karst can have a strong influence on runoff modelling. In our study the influence of karst on the catchment scale is relatively small which is reflected on the reasonable well simulated hydrographs.

**P5 L8:** Was it really necessary with 3 years spin up time to establish state variables in HBV.... it is commonly done over much shorter time. This need some explanation. Usually one prefer to have as long calibration period as possible to catch as many different met variations and combinations as possible.

**Reply:** For many cases a warm-up period of one year will be sufficient for HBV-light. However, longer warm-up periods improve the conditions of the state variables and don't have a negative effect on the simulations. We used 2 years and 9 months for warming up because it allowed us the most optimal use of the time series with two equally long modelling time periods covering full hydrological years. The runoff time series lasted from January 1982 to December 2009.

**P5 L9:** This is not clear. Did you use the first period for calibration 84-96 where 84 to 87 was spin up time, and 97-09 for validation and 97-00 as spin up... or

**Reply:** For the simulation period of October 84 - October 96 the warming up was from January 82 – September 84. For the simulation period of October 97 - October 09 the warming up was from January 95 – September 97.

We will change "A three-year calibration period. . ." to "A three-year warm-up period. . .", because this might have been confusing.

**P5 L24:** consist of one single SFC that incoprorate 13 SFC.. I do not understand what you are saying here.

**Reply:** We will adapt the sentence to make it more clear. It means that we defined an objective function that consists of one single SFC (ISinlge). Since we have 13 ecologically relevant SFCs in our study catchments, we also have 13 versions of that new objective function (ISinlge).

**P6 L1:** ...are you not mixing the term objective funtion and SFC here... if not this is very unclearly formulated...

**P6 L2:** I think I understand what you try to say....but is this formulation good? Rewrite to make it clearer and separate between the function and the SFC's

**Reply:** The two comments above relate to each other and thus we answer them in one paragraph:

We agree that the terms SFCs and objective function are not properly used. We will adapt the sentence so that it becomes clear that we selected SFCs that resulted in robust and informative estimates when used as objective function. The two selection criteria of robustness and information value should help to define an IMulti that will be relatively robust and informative.

**P6 L5:** Combine evaluation and calibration chp these are so cloes that they should be in same chp. And the sfc discussion should be with sfc description.

**Reply:** Model calibration and evaluation seem to be close, but there are some important differences in the criteria we use or how we use criteria (see comment P19 L4 in detailed comments RC 1 about tables and figures). Thus, we prefer to keep the two parts separated to emphasize the difference and reduce the potential for confusion.

**P6 L6:** ? do you try to say that you use the five criterion's in table 2 and Mare. You have already stated how SFC's are included in thes and do not need to say so again.

**Reply:** We only use the criteria in table 3 for model evaluation (see comment P19 L4 in detailed comments RC 1 about tables and figures). We will change the sentence from "...was evaluated by means of SFCs, Reff and mean relative error (MARE)." to "...was evaluated by means of normalized SFC error, Reff and mean relative error (MARE)." to emphasize the different use of the SFCs in calibration and evaluation.

**P6 L9:** Why choosing the median parameter set? And what is median parameter set? Is it not normal to use the optimal parameter set? I do not understand the purpose of the interpretation using a median parameter set? Unless clearly described later this must be explained here.

**Reply:** The 100 calibrations done with each objective function result in 100 optimized parameter sets. These hundred different parameter sets lead to very similar model performance in calibration, which is a common observation usually referred to as equifinality (Beven and Freer, 2001). The equifinality concept rejects the idea of one single best parameter set. We calculated the median model performance of all 100 parameter sets in calibration and validation to not select the best, but rather a representative value from the efficiency distribution.

Beven, K., and Freer, J.: Equifinality, data assimilation, and uncertainty estimation in mechanistic modelling of complex environmental systems using the GLUE methodology, J. Hydrol. 249, 11–29, doi:10.1016/S0022-1694(01)00421-8, 2001.

**P6 L10:** Should be in the chp where SFC are described.

**Reply:** Ok, we will move that sentence to chapter 2.2.

**P6 L22:** Did you not optimize on both these criterions...if not that must be mead clear earlier. If you did...then it should result in 26 parametersets...

**Reply:** Correct, it's two times 13 resulting in totally 26.

**P6 L22:** I assume the calibration results in a optimal parameterset given the criterion used. And not in a simulation...

**Reply:** We will adapt the sentence to state it more precise.

**P6 L24:** But what about variability? Is it not better to have high variability and some good model simulations than low variability around poor simulations (as I assume a low score indicates)

**P6 L24:** here you say you uses both Isingle and the combined I signle and I reff ...my previous comment asks about this...

**Reply:** The two comments above relate to each other and thus we answer them in one paragraph:

The optimal value for the normalized SFC error is 0 (see Table 3). In the text we describe the variability of the error value and its magnitude for the objective functions Reff, ISingle and ISingle_Reff. The best situation would be a low variability at a low error magnitude. Low variability at a high error magnitude would indicate that the related objective function is not suitable for model calibration aiming at the respective SFC. High error variability combined with some low error magnitudes would indicate that certain SFCs used as objective function (ISingle) can lead to good estimates. But it also means that not any SFCs used in ISingle results in good SFC estimates. This fact supports our main conclusion that SFCs should preferably be estimated from targeted runoff model calibration.

**P6 L27:** In general or for TA1?

**Reply:** The description refers to the results of TA1 which was selected as a representative example. But overall a similar pattern can be seen for all SFCs.

**P6 L28:** illustrated by...

**Reply:** Will be added.

**P7 L1:** Again...what is median simulation?

**Reply:** For each SFC we get 13 normalized errors for the objective functions ISingle and ISingle_Reff (as shown in Figure 3). The median of these 13 normalized SFC errors is displayed in Figure 4 for each SFC.

**P7 L1:** Reff of <0.1 is very poor and telle me that there are something substatially wrong. Do I misunderstand? Reff should be higher than at least 0.5 before you can say you have representative data and higher than at least 0.7 before you can say that you are able to model a catchment satisfactory.

**Reply:** The Nash-Sutcliffe efficiency is not displayed in Figure 4. For more discussion on the Nash-Sutcliffe efficiency please see comment 1 in RC 1.

**P7 L6:** The Reff is very low for all catchments it seems like. only one above 0.5. This tells me the opposite of you comment here! Why is Reff so low. Poor precip data og runoff data or??? Is a acheived criterion below 0.2 good agreement with obeserved? In that case you have to explain how.

**Reply:** We will add some text in the results part of the manuscript to clarify the confusion regarding the Nash-Sutcliffe efficiency and to avoid any misunderstanding. In figure 5, the y-axis of Reff (right axis) ranges from 0 on top to 1 at the bottom. The closer to 1 (thus to the bottom), the better is the Nash-Sutcliffe efficiency. For all 5 types of objective functions the value of Nash-Sutcliffe is above 0.5.

---

## Author Comment (AC3) · 7 Dec 2016

Dear Björn Guse,

Thank you for your efforts with our manuscript. We greatly appreciated the comments, which will help us to improve the manuscript. Please see below our detailed response to each of the comments.

Best regards,

Sandra Pool and Co-Authors

**Major comments RC 3**

[Figure]

**Comment 1:** In its current state, the article is in my opinion mainly related to hydrology (also by the selection of the journal). At several parts the authors emphasized its ecological importance (e.g. Page 1, Line 10, P.2, L.1-2, P.11, L.14-16). However, I do not really see a connection to ecology. Thus, either the article has to be fully focused on hydrology or it is required to emphasize its ecological relevance

**Reply 1:** We agree that our modelling approach for estimating SFCs is a typical hydrological application. However, the motivation for our study is strongly related to ecohydrology because of its focus on the simulation of ecologically relevant SFCs of our study catchments (chapter 2.2). While many of our SFCs are also the focus of other ecohydrological studies, they are untypical for purely hydrological modelling studies where signatures such as segments of the FDC or runoff ratio are of interest. Given the importance of SFCs in ecology we find it important that the hydrological community addresses the issue how to compute these values for ecological studies and applications.

**Comment 2:** In this article, the Nash-Sutcliffe Efficiency is used as an example for a traditional calibration approach. However, in recent studies the use of several performance measures related to different parts of the hydrological system is recommended. In particular the use of typical performance metrics such as Reff or Kling-Gupta-Efficiency (KGE) in combination with signature measures is recommened (see eg. Van Werkhoven et al., 2009). Thus, I think that a comparison only with the Reff is not sufficient. I recommend to use two or three performance measures such as PBIAS or KGE to show that SFCs also outperforms a calibration approach based on NSE in combination with PBIAS (or other performance measures).

**Reply 2:** We agree that combined objective functions based on e.g. Nash-Sutcliffe efficiency (Reff) and volume error are widely used for runoff model calibration. We actually used such combined objective functions (that are not based on SFCs) in our previous study (Vis et al., 2015) to estimate the same SFCs of the same catchments as in the current study. The average SFCs error (percent error) in calibration was lowest for

calibrations with Reff (error of -4.9 %), followed by calibrations with objective functions based on a) Reff, LogReff and volume error (error of -5.1 %), b) Reff and volume error (error of -5.6 %) and c) Reff, MARE, spearman rank correlation and volume error (error of -6.1 %). Some other combined objective functions were also tested, but resulted in clearly poorer SFC estimates than the ones listed above. We therefore decided to use Reff as a benchmark in the current study. We plan to add some information about the results of the preceding study in the introduction of our manuscript.

Vis, M., Knight, R., Pool, S., Wolfe, W., and Seibert, J.: Model calibration criteria for estimating ecological flow characteristics, Water, 7, 2358–2381, doi:10.3390/w7052358, 2015.

**Comment 3:** Moreover, it needs at least to be discussed how a calibration approach based on these SFCs is related to recent studies using hydrological signatures such as segments of the FDC (see Yilmaz et al., 2008, Pfannerstill et al., 2014).

**Reply 3:** Thank you for making us aware of the study from Yilmaz et al. (2008) which we will mention in the introduction in addition to the study of Pfannerstill et al. (2014). These two studies use hydrological signatures to better predict the overall hydrograph and thus their main goal slightly differs from ours. We plan to briefly compare our results with theirs in the discussion.

**Comment 4:** P. 1, L. 24-27: I do not fully agree with this statement. There are different ways of how to calibrate a discharge time series. Certainly there are studies which are focused on certain parts such as on high or low flows. However, other studies aim to represent the whole hydrological system at best without neglecting or emphasizing certain parts. In the way towards a good representation of the hydrological system, the latter one should be the general goal and a strong focus on certain parts of the hydrological system should be a specific case.

**Reply 4:** We agree that some general agreement often is the calibration goal. Our statements aims at the fact that even with such a general agreement, specific aspects

might be poorly simulated as we have shown in the preceding study (see Vis et al., 2015).

**Comment 5:** P.2, L. 21: Here, SFCs are defined "as equivalent to hydrological signatures". In this case, I do not understand the use of SFC. It is stated before that hydrological signatures are a more common term in hydrology. It is certainly required to justify why you used the term SFC since I do not really see the strong relationship to ecology.

**Reply 5:** As described in the reply of comment 1, the selection of the SFCs is motivated by their ecological relevance in the study catchments. While both terms, hydrological signature and SFC, describe characteristics of the hydrograph, only the term SFCs makes a distinct connection to ecology. The term SFC has also been used for many years, whereas the term hydrological signature has been introduced more recently.

**Comment 6:** I think that the article would benefit from a more detailed interpretation of the results. For example on P. 7. L. 12-15; P.8, L.17-18: Can you explain these results or more specifically the behaviour of these SFCs?

**Reply 6:** We will discuss the behavior of the indicated exceptions in the revised manuscript.

**Comment 7:** It could be interesting to analyze the relationship/correlation between the SFCs.

**Reply 7:** We did a Spearman rank correlation test for the 13 SFCs using all 25 catchments. We did the correlation test for both modelling time periods and figure 1 (in this document) shows the average correlation value of the two time periods. The correlation values of the two time periods were similar.

**Comment 8:** The combination of different metrics might outperform in general single-metric approaches. However, the more metrics are included, the more a trade-off might occur and the equifinality problem arises. In this context, can you give a recommendation for a good number of required SFCs? In the best case, a systematic way of how to select the best SFCs will be provided. Even though when I expect that it is difficult to find a precise number, it is worth discussing this point.

**Reply 8:** Adding more criteria into a combined objective function usually rather decreases than increases the equifinality issue according to our experience and other studies, although you are right that also an increase because of compromise situations might be possible.

We systematically selected SFCs for a multi-objective function based on the criteria of information value and robustness. Our results indicate the importance of jointly evaluating both criteria for the selection of SFCs for a multi-objective function. However, we cannot give a minimum number of SFCs required for such an objective function, because this will depend on the type and combination of SFCs one is interested in. We could show that the four SFCs used for the multi-objective function preserved similar hydrograph characteristics as the Nash-Sutcliffe efficiency. How much value additional SFCs have for the representation of the hydrograph characteristics would have to be evaluated using many more than 13 SFCs and eventually using synthetic data to also see the effect of redundant information (see discussion chapter 4.3).

**Comment 9:** The figures 3-8 are very similar (at least visually). I think that the article would benefit from emphasizing the relevance of each figure. It is partly difficult to differentiate them. Maybe you can also thinking about reducing the number of figures to improve the overall message. For example, the figures 6 and 8 have almost the same figure caption. To summarize this point, it is easier to detect the whole message in the case of a more distinct presentation of the results. One example for this is the figure 9 which can be clearly distinguished from the other figures. These results are easier to understand.

**Reply 9:** We agree that some figures are similar and maybe hard to distinguish. We will try to change or reduce the most similar figures to present the results in a more

interesting or intuitive way in the revised manuscript.

**Comment 10:** P. 8, L.12-13: Could you specify how you can here differentiate between error dependence on time period or objective function?

**Reply 10:** We were interested in finding systematic patterns in the error magnitude (absolute value of normalized SFC error) of the SFCs. For some SFCs (e.g. TL1) the error magnitude could be considerably higher in one modelling time period than in the other one. We described the error of such SFCs as being time depended. Some other SFCs had clearly higher error magnitudes when calibrated on a certain objective function (e.g. MH10). The estimation accuracy of such SFCs was considered as being dependent on the objective function.

**Comment 11:** P.8, L.14-15: I do not understand this statement that the SFCs are neither related to flow components nor to flow conditions. Hydrological signatures (as an equivalent term) are known to be of special importance to explain the hydrological behaviour. Thus, what can we learn from using these SFCs in terms of the hydrological behavior in the catchment. And how is this related to the general idea of the hydrological signatures?

**Reply 11:** We are sorry for the confusion this statement evoked and will make the sentence more clear. We wanted to say that we could not relate the estimation accuracy (error magnitude, spread of error magnitude and dependency of error magnitude on modelling time period/ objective function (see comment 10)) of the analyzed SFCs to the flow components or flow conditions they belong to. E.g. we cannot say that all high-flow related SFCs had very low estimation accuracies.

**Comment 12:** P.8, L.26 to P. 9, L.21: I agree with this part which is clearly understandable, but certainly also not surprising. It is mostly existing knowledge of hydrological modelling. What can we learn here except of using several and different metrics. I recommend to shorten this part and emphasize the most important points from this study. In contrast, I really like to following passage (P. 9, L.21-31). Please also discuss the

impact of a SFC-based calibration for the process representation. Can you state that the hydrological system is overall better represented by using several SFCs? Please discuss the benefit of optimizing one specific SFC. This leads to a modelled hydrograph which is able to represent a very specific condition but probably not the overall hydrograph. This implies that the part of the hydrological system which is not in the focus of this SFC is probably not adequately considered. This might be of particular relevance when using very specific SFCs such as MA26.

**Reply 12:** Thank you for this helpful comment. We will adapt the discussion part you mentioned focusing on your proposed aspect of how the calibration with SFCs affects process representations:

The benefit of optimizing one specific SFCs lies in the relatively accurate estimation of the respective SFC compared to a calibration with Reff or a multi-SFC objective function. Model calibration on one single SFC clearly emphasizes the hydrograph aspects of the selected SFC which can negatively affect other hydrograph aspects. This implies that calibrations with ISingle can lead to poor model performance for SFCs not included in the objective function (figure 4). E.g. calibrating on the frequency of moderate floods (FH6) leads to poor model efficiencies for baseflow (ML20) (figure 7b) which indicates that the representation of the main runoff processes can suffer by SFC-specific model calibration. From the fact that a calibration with Reff and a calibration with multiple SFCs lead to comparable SFC estimates we infer that the main hydrological processes of the catchments are similarly well represented with the two approaches. We assume that these two calibration criteria result in a better process representation than the calibration with a single SFC, because they outperform the calibration with ISingle for those SFCs not included in ISingle.

**Minor comments RC 3:**

**P.1, L. 12:** maybe "optimization" instead of "minimization or maximization".

**Reply:** We will do the replacement as suggested.

**P.1., L.16:** Are these over- and underestimations a general aspect of these SFCs or are they case-specific?

**Reply:** These results represent the general tendency of the model and the results from a specific objective function and/or modelling time period can deviate from these general conclusions (figure 9).

**P.2, L.16-19 and L. 30:** I recommend to include the study from Yilmaz et al. (2008).

**Reply:** Thank you for this suggestion. We will add the study of Yilmaz et al. (2008) to the mentioned part in the introduction.

**P.2, L. 34:** The meaning of "esoteric and subtle aspects of the flow regime" is unclear.

**Reply:** Good point, we will adapt the sentence.

**P.3, L. 19:** Please think about renaming the section to "Methods and materials", since a catchment is not a method.

**Reply:** We will change this title as suggested.

**P.5, L.31:** Why you have used 0.2 and 0.25 as weights?

**Reply:** IMulti consists of four SFCs that we wanted to weight equally which leads to weights of 0.25 for each SFCs. These four SFCs were combined with the Nash-Sutcliffe efficiency (IMulti_Reff) and each of the components was again assigned the same weight (0.2). We weighted all components of IMulti and IMulti_Reff equally because there was no convincing reason to weight one/some SFC(s) more than others.

**P.6, L. 9:** Could you specify "median parameter set"?

**Reply:** This sentence should say ". . ., the median model efficiency of each catchment was selected." We will change this sentence.

**P. 11, L.15:** Could you specify "later application of simulated SFCs related to flow

alteration – ecosystem change relationships". This aspect was up to now neither in the focus of the article nor emphasized as an overall aim.

**Reply:** This statement refers to the first paragraph of the introduction where we motivate the need for accurate SFC estimates by giving the example of flow alteration – ecosystem change relationships used for sustainable flow management. We will adapt the sentence in the discussion to make this connection more clear.

**Table 1:** TA1: runoff with two f

**Reply:** We will add the missing f.

**Table 1:** Why you have named the SFCs FH6 and FH7 and not FH3 and FH7.

**Reply:** We agree that the abbreviation can be confusing, but decided to follow the abbreviations used in previous publications and in the EflowStats R-package that was used for the calculation of the SFCs. The same abbreviations are commonly used in many studies (see e.g. Olden and Poff, 2003).

Olden, J. D., Poff, N. L.: Redundancy and the choice of hydrologic indices for characterizing streamflow regimes, River Research and Applications, 19(2), 101-121, doi: 10.1002/rra.700, 2003.

**Fig. 5:** Can you explain the outlier TL1?

**Reply:** We actually already discussed the low estimation accuracy of TL1 in the discussion (last paragraph of chapter 4.1).

[Figure]

| | Mean-flow | | | | Low-flow | | | | High-flow | | | | |
|------|------|------|------|------|------|------|------|------|------|------|------|------|------|
| | ma41 | ma26 | ra7 | ta1 | e85 | fl2 | tl1 | ml20 | mh10 | dh13 | dh16 | fh6 | fh7 |
| ma41 | - | | | | | | | | | | | | |
| ma26 | -0.529 | - | | | | | | | | | | | |
| ra7 | -0.449 | 0.843 | - | | | | | | | | | | |
| ta1 | -0.844 | 0.653 | 0.632 | - | | | | | | | | | |
| e85 | 0.803 | -0.811 | -0.801 | -0.852 | - | | | | | | | | |
| fl2 | 0.213 | -0.506 | -0.697 | -0.395 | 0.500 | - | | | | | | | |
| tl1 | -0.393 | 0.130 | 0.120 | 0.377 | -0.165 | -0.097 | - | | | | | | |
| ml20 | 0.546 | -0.894 | -0.962 | -0.736 | 0.869 | 0.688 | -0.148 | - | | | | | |
| mh10 | 0.840 | -0.425 | -0.402 | -0.725 | 0.689 | 0.198 | -0.475 | 0.463 | - | | | | |
| dh13 | -0.650 | 0.864 | 0.867 | 0.804 | -0.918 | -0.600 | 0.120 | -0.945 | -0.540 | - | | | |
| dh16 | 0.513 | -0.646 | -0.722 | -0.642 | 0.678 | 0.600 | -0.319 | 0.750 | 0.545 | -0.650 | - | | |
| fh6 | -0.484 | 0.865 | 0.902 | 0.632 | -0.796 | -0.662 | 0.103 | -0.934 | -0.364 | 0.876 | -0.677 | - | |
| fh7 | -0.614 | 0.884 | 0.895 | 0.762 | -0.901 | -0.609 | 0.112 | -0.947 | -0.503 | 0.966 | -0.666 | 0.917 | - |

**Fig. 1.** Spearman rank correlation coefficients for the 13 SFCs.

---

## Editor Comment (EC1) · D. Solomatine (Editor) · 19 Dec 2016

A good discussion on an old subject - which indeed still needs attention. It can be also useful to remind young authors that "blind calibration" is worse than no calibration at all, and that "if you optimise something, first think well what is the objective function".

It can be seen that the authors appreciate the comments, and the (improved) version of the paper is expected.

Season's greetings to all!

* * * * *
* * *

---

## Author Response (AR1)

| | |
|---|---|
| **Journal:** | Hydrology and Earth System Sciences (HESS) |
| **Title:** | Streamflow characteristics from modelled runoff time series - importance of calibration criteria selection |
| **Manuscript:** | hess-2016-546 |
| **Authors:** | Sandra Pool, Marc J. P. Vis, Rodney R. Knight, and Jan Seibert |

Dear editor,

We thank you for your efforts with our manuscript. The two reviewers provided valuable comments on our manuscript, which helped us to improve the quality of our manuscript. Below, we reply to each of the comments from the reviewers and indicate the changes that have been done accordingly (marked with blue color). We also did a few minor edits that were not suggested by the reviewers. References to pages (P), lines (L), chapters, figures and tables refer the track-changed revised version of our manuscript.

On the behalf of all Co-Authors,
Yours sincerely,
Sandra Pool

**Response from the editor**

A good discussion on an old subject - which indeed still needs attention. It can be also useful to remind young authors that "blind calibration" is worse than no calibration at all, and that "if you optimise something, first think well what is the objective function".
It can be seen that the authors appreciate the comments, and the (improved) version of the paper is expected.
Season's greetings to all!

**Review 1: comments, response and modifications**

Dear reviewer,

Thank you for your efforts with our manuscript. We greatly appreciated the comments, which helped us to improve the manuscript. Please see below our detailed response to each of the comments.

Best regards,

Sandra Pool and Co-Authors

**General comments RC 1**
**Comment 1:** It is a very good idea and a well know idea, that the models should be calibrated using criterions relevant to the purpose of the model and simulations. The criterions here are many but seems relevant to ecological studies. The relevance of the choosen SFC should be argued for in relevance to the purpose. Here are maybe to many SFC and the it becomes difficult to keep track of them. Are they correlated, are they in contradiction (to fulfill one means that another suffer) etc. It might be an idea to pick the most important for the kind of studies the model should be used in and discuss this closer.
Reply 1: We agree that we used many SFCs. There are more than 150 different SFCs used in ecohydrological studies and many of them are correlated or redundant (Olden and Poff, 2003). In our study we use 13 SFCs that have been shown to be the most relevant ones for the ecological integrity of the study catchments in previous studies (see chapter 2.2 for the motivation of the selected SFCs). We

are aware that some of the selected 13 SFCs are correlated, e.g. FH6 and FH7 (Table 1 of this response), but still decided to use these ecologically most relevant SFCs in our study. As you suggest, selecting the most important SFC would allow a deeper analysis but also reduce the potential for general conclusions. Our 13 ecologically relevant SFCs have the advantage of representing different flow components and flow conditions.

Calibrating the model for one SFC can indeed have a negative effect on the estimation accuracy of another SFC (e.g. calibrating for FH6 (high flow) could negatively affects ML20 (low flow)) (see also our discussion in chapter 4.1). This effect seems to be inevitable in runoff modelling because model calibration is a trade-off where usually no perfect parameterization can be found due to different uncertainty sources. In case of a perfect situation with no model, parameter and data uncertainty, all SFCs could be perfectly estimated with the same runoff simulation.

**Comment 2:** But before it is relevant to discuss other criterions and SFC used for calibration and how well these can be recreated you need to demontrate that teh model is able to reproduce observed flow to a certain degree. I can not see that this is the case here. As long as this is not the case and is demontrated the remaining work becomes irrelevant. If the model is not able to get an Reff higher than 0.7 the interesting discussion is then if it is able to or relevant to use for simulation of other SFC.

**Reply 2:** Thank you for this useful comment. We agree that it is important to know that HBV can reproduce observed runoff to an acceptable degree. Based on your feedback we realized that we didn't clearly formulate this, and especially the presentation and labels of some of the figures were misleading. We adapted the axis labels of Fig. 3, 5, and 7 and simplified the axis of Fig. 4. These graphs probably caused most of the misunderstanding. We also added a paragraph to the results part with the information that HBV-light model was capable of reproducing the observed runoff for the study catchments reasonably well when calibrated on the Nash-Sutcliffe efficiency ($R_{eff}$). Model calibration on $R_{eff}$ resulted in $R_{eff}$ values between 0.68 and 0.89 with a median of 0.79 in calibration. Model performance in calibration was above 0.7 for all except one catchment. The corresponding $R_{eff}$ values in validation ranged from 0.62–0.86, whereby the median model efficiency was 0.77 and efficiencies were larger than 0.7 for 21 of 25 catchments." This means that it is in principle possible to get good performances in terms of $R_{eff}$, or, in other words, poorer performances are a result of the respective calibration criteria rather than an inappropriate model (structure).

**Modification:** P25-26 (Figure 3), P28-29 (Figure 4), P29-30 (Figure 5), P31 (Figure 7), P7 L11-13 (HBV calibration efficiency)

**Comment 3:** It might be that you are able to acheive good simulations in this catchment. I that case show this also by showing hydrographs. If you are not able to achieve a Reff up to 0.7 you need to disuss why first and then move to other STC that you can argue for the model is able to recreate in spite of a poor Reff. If the Reff is OK, but as you initially says is not enough for some studies, you can compare calibration with other criterions againt Reff and each other. I beileve that it is the latter that you try to do, but as long as I as a reader am not able to see that your model and data actually are representative or good enough to reporduce the observed the discussion becomes not relevant or interesting.

**Reply 3:** Yes, we indeed calibrated the model with different objective functions and then compared the resulting model efficiencies.

As discussed in our reply to comment 2 of RC1, HBV was able to reasonably well reproduce the hydrographs of the 25 catchments. We decided to only report the Nash-Sutcliffe efficiencies and not to show the hydrographs. The hydrographs of the complete simulation period from 1984-2009 do not provide much information when shown in one graph (low visibility of information) and the details of one year might not be representative. However, the Nash-Sutcliffe efficiency is an integrated measure of the hydrograph fit over all years. We therefore decided to only report Nash-Sutcliffe efficiencies and not to show hydrographs. We also assume that the request for showing hydrographs came most out of the misunderstanding that the model would be very poor.

**Comment 4:** So before revising this deeper several clification have to be made initially. But the topic is very intersting and relevant so I hope you are able to structure this in a way that make many readers interested and enlightened.

**Reply 4:** We addressed your more detailed comments in the parts below (see detailed comments RC 1)

**General comments RC 2**

**Comment 1:** You mention in generalm terms that high peaks and low flow is not well enough simulated. Could this and other features the "good Reff model" does not capture be illustrated in the introduction as a background for the study and choice of additional criterions?

**Reply 1:** This is a good idea. We extended the introduction by a part about the drawback of some commonly used calibration criteria for low and high flow related SFCs. In our previous study model calibrations with widely used objective functions such as Nash-Sutcliffe efficiency, volume error, MARE, etc. resulted in the underestimation of high-flow related SFCs by 13-33% and overestimation of low-flow related SFCs by 4-23%. In the current manuscript we analyze the estimation accuracy of the same SFCs when calibrated with the new approach that explicitly takes into account the SFCs of interest.

**Modification:** P3 L2-8

**Comment 2:** The paper would also benefit of a short argumentation for the choice of SFC as calibration criterions. And also how these are calculated. Ieff does not tell med how you find the value for the criterions and the table does not say so either.

**Reply 2:** Table 1 in the manuscript indeed only gives a description of the SFCs used in this study. For the calculation of the SFCs the EflowStats R-Package from the USGS (chapter 2.2 in the revised manuscript) was used. More details about the exact calculation can be found in the R scripts and the corresponding documentation. The R-package is freely available and we recommend reader who are interested in the exact mathematical formulations to check the R-package for detailed information on the calculation of the SFCs. For most readers, however, we argue that the idea of the indices as given in Table 1 is sufficient, especially since the 13 SFCs used in our study are commonly used SFCs and can be found in many ecohydrological studies. We further discuss the choice of the SFCs in comment 1 in RC 1.

**Detailed comments RC 1: tables and figures**

**P17 L1:** Table 1- But how are these used. For instance how do you evaluate simulated against observed for TA1? This should be explained for all SFC's.

**Reply:** We use SFCs for model calibration and validation, whereby simulated SFC values are always evaluated against observed SFC value. The exact definition of the objective function used for calibration can be found in Table 2 ($I_{Single}$), whereas the performance measure used for validation is given in Table 3 (nSFC). The same equations are used for comparison of simulated and observed values for all SFCs.

**P17 L3-30:** Table 1 - smaller comments on the description of SFCs

**Reply:** Thank you for these comments on the description of the SFCs. We adapted the text in the table according to your input.

**Modification:** P19 (Table 1)

**P19 L4:** Table 3 - It is not clear for me why you need two tables for evaluation/calibration criterions. I assume you use them all in different calibrations....and then also the same in evaluating the performance? From fig 2 I get the idea. But this should be better described in the text. Reff is used both in calibration and evaluation. Describe why.

**Reply:** We decided to have two tables to make clear which criteria are used for model calibration and which criteria are used for model evaluation. We also used a separate chapter for the description of the calibration and evaluation criteria (chapter 2.4.2 and 2.4.3) to make the difference more clear. Nash-Sutcliffe efficiency is the only criteria used in both calibration and evaluation. SFCs are used for

model calibration and evaluation, but the exact definition is slightly different. For model evaluation the SFCs had to be normalized to make the results comparable (see chapter 2.4.3). The combined objective functions $I_{Single\_Reff}$, $I_{Multi}$, $I_{Multi\_Reff}$ are only used in model calibration, whereas MARE is only used in model evaluation. We think it is helpful for the reader to keep those two tables separated, especially because the use of the SFCs is different.

As you mentioned, we use the Nash-Sutcliffe efficiency in model calibration and evaluation. The reason for that is that the Nash-Sutcliffe efficiency is an established measure used in many modelling studies in the same way as we do. It shows how well the model is reproducing the criteria it was calibrated on in an independent time period. It is also a measure of how well the general shape of the hydrograph is simulated, although with a focus on peaks. The use of Nash-Sutcliffe in our model evaluation is useful for the comparison to other studies.

**P22 L4:** Figure 3 - You refer too this figure as an illustration of variability of model perfomance...and her say it is comaprison between calibration and validation period. This is no consistent. You also say it is about A1 but shows all SFC. I do not get it. Neither that you have stablke and very poor Reff...

**Reply:** We apologize for the confusion this figure evoked. We admit that the axis labels were not well selected and that they were misleading. We changed the axis labels to make this plot better readable. For example the axis label "$R_{eff}$" was changed to "Normalized TA1 error for model calibration with $R_{eff}$". Additionally, we changed the figure to a 3-D graph which on one hand allowed us to show all the relevant information in one single plot and on the other hand reduced the number of subplots from 6 to 2. For clarification we shortly describe the figure here: The figure shows the normalized TA1 error when calibrated with $R_{eff}$, $I_{Single}$ and $I_{Single\_Reff}$. When calibrating the model based on the Nash-Sutcliffe efficiency, we get one normalized TA1 error for each modelling time period. This results in two values of the normalized TA1 error displayed at the $R_{eff}$ -axis. Thus, the value of approximately 0.1 at the $R_{eff}$ -axis is the normalized TA1 error and not a Nash-Sutcliffe efficiency. Since we have 13 different SFCs that we use for objective functions, we get 13 normalized TA1 error values from calibrations with $I_{Sinlge}$ or $I_{Single\_Reff}$. These are the 13 values displayed at the $I_{Sinlge}$-axis or the $I_{Single\_Reff}$-axis. Each of the 13 different objective functions are colored based on the SFCs it is based on.

We hope that with the change of the axis labels, the plot type and the figure caption it becomes clearer that we want to illustrate the variability of model performance and that we are not comparing calibration and validation.

**Modification:** P25-26 (Figure 3)

**Detailed comments RC 1: text**

**P1 L10:** What is the purpose? It does not come clearly forward here. Is it to simulate streamflow characteristic in ungauged basin. In that case is the simulation results relevant for ungauged basins? Is the test sites ungauged, is it split samle testing? Or is it ordinary calibrated models for guaged basin. In cas of latter, how representativ is this test then for ungauged basins?

**P1 L14:** I assume it's here it is satde what the actual purpose is. But could the strategy and the purpose be stated clearer. To a person only reading the abstract to consider if this is interesting I do not think this tells enough.

**P1 L19:** For ungauged basisn estimates or in general?

**Reply:** The three comments above all address the abstract, so we answer them in one paragraph:

We agree that the abstract needed to be improved to clearly state the purpose and the method of this study. We therefore rewrote the abstract taking the comments into consideration. In our study we only work with gauged catchments. The aim was to test whether the consideration of SFCs in model calibration improves the estimation accuracy of SFCs compared to more traditional calibration approaches using e.g. the Nash-Sutcliffe efficiency. Our results help to improve model calibration for estimating SFCs, which is of great importance for a subsequent regionalization of SFCs for ungauged catchments. The ultimate aim is therefore to have improved model calibration approaches for the regionalization of runoff to ungauged catchments. For gauged catchments we don't need to model runoff for the estimation of SFCs because they can be calculated form the observed data.

**Modification:** P1 L11-23

**P1 L27:** I can not see many other applications for runoff simulations than recreating streamflow characteristics. so in that sense this sentence does not make sense. But if it is ecological SFC then I can see this is referred to as spesific SFC. So it migth be that ecological shoudl be added. Here and other places in the document.
**Reply:** We deleted this sentence and added the term "ecologically relevant" to the subsequent sentence.
**Modification:** P2 L3-4

**P3 L9:** But initially you stated that this is what is the challange...."Ecologically relevant streamflow characteristics (SFCs) of ungauged catchments are often estimated from simulated runoff of hydrologic models." ... and the rest of the paper is about gauged basin where this challange is substantially lower... this confuses the reader as this is two very different challanges. You can discuss this but make clear already in abstract that this paper is about gauged catchments.
**Reply:** True, we tried to make sure in the abstract that our study is about gauged catchments (see comment P1 L10 in detailed comments RC 1 about text).

**P3 L16:** Other than?
**Reply:** We replaced "… or more other SFCs?" by "... or multiple SFCs?"
**Modification:** P4 L5

**P3 L17:** Spesific SFC is included also in traditional calibration (max and mean is spesific SFC). But maybe not those intersting for a spesific purpose. Understand the point, but is not clearly formulated.
**Reply:** We replaced "…and those where specific SFCs are included?" by "… and those where the SFCs of interest are included?"
**Modification:** P4 L6-7

**P4 L1:** Is usually a challenge if water drain out uncontrolled. Is this a problem in this catchment?
**Reply:** We agree that karst can have a strong influence on runoff modelling. In our study the influence of karst on the catchment scale is relatively small which is reflected on the reasonable well simulated hydrographs.

**P5 L8:** Was it really necessary with 3 years spin up time to establish state variables in HBV.... it is commonly done over much shorter time. This need some explanation. Usually one prefer to have as long calibration period as possible to catch as many different met variations and combinations as possible.
**Reply:** For many cases a warm-up period of one year will be sufficient for HBV-light. However, longer warm-up periods ensure that the conditions of all the state variables are really in equilibrium. Besides requiring data, a longer warming-up period does not have a negative effect on the simulations and for certain parameterizations the typical one-year warming-up actually is too short if one looks in detail at the groundwater storage. We used 2 years and 9 months for warming up because it allowed an optimal use of the time series with two equally long modelling time periods covering full hydrological years. The runoff time series lasted from January 1982 to December 2009.

**P5 L9:** This is not clear. Did you use the first period for calibration 84-96 where 84 to 87 was spin up time, and 97-09 for validation and 97-00 as spin up... or
**Reply:** For the simulation period of October 84 - October 96 the warming up was from January 82 – September 84. For the simulation period of October 97 - October 09 the warming up was from January 95 – September 97.
We added the dates for the warm-up periods and changed "A three-year calibration period…" to "A warm-up period…", because this might have been confusing.
**Modification:** P5 L28-29

**P5 L24:** consist of one single SFC that incoporate 13 SFC.. I do not understand what you are saying here.
**Reply:** We adapted this part to make it more clear. It means that we defined an objective function that consists of one single SFC ($I_{\text{Sinlge}}$). Since we have 13 ecologically relevant SFCs in our study catchments, we also have 13 versions of that new objective function ($I_{\text{Sinlge}}$).
**Modification:** P6 L13-15

**P6 L1:** ...are you not mixing the term objective funtion and SFC here... if not this is very unclearly formulated...
**P6 L2:** I think I understand what you try to say....but is this formulation good? Rewrite to make it clearer and separate between the function and the SFC's
**Reply:** The two comments above relate to each other and thus we answer them together:
We agree that the terms SFCs and objective function are not properly used. We adapted the sentence so that it becomes clear that we selected SFCs that resulted in robust and informative estimates when used as objective function. The two selection criteria of robustness and information value should help to define an $I_{\text{Multi}}$ that will be relatively robust and informative.
**Modification:** P6 L23-25

**P6 L5:** Combine evaluation and calibration chp these are so cloes that they should be in same chp. And the sfc discussion should be with sfc description.
**Reply:** Model calibration and evaluation seem to be close, but there are some important differences in the criteria we use or how we use criteria (see comment P19 L4 in detailed comments RC 1 about tables and figures). Thus, we prefer to keep the two parts separated to emphasize the difference and reduce the potential for confusion.

**P6 L6:** ? do you try to say that you use the five criterion's in table 2 and Mare. You have already stated how SFC's are included in thes and do not need to say so again.
**Reply:** We only use the criteria in Table 3 for model evaluation (see comment P19 L4 in detailed comments RC 1 about tables and figures). We changed the sentence from "…was evaluated by means of SFCs, $R_{\text{eff}}$ and mean relative error (MARE)." to "…was evaluated by means of normalized SFC error, $R_{\text{eff}}$ and mean absolute relative error (MARE)." to emphasize the different use of the SFCs in calibration and evaluation.
**Modification:** P6 L29

**P6 L9:** Why choosing the median parameter set? And what is median parameter set? Is it not normal to use the optimal parameter set? I do not understand the purpose of the interpretation using a median parameter set? Unless clearly described later this must be explained here.
**Reply:** In this sentence we wanted to refer to the median efficiency and not the median parameter set. We adapted the term accordingly. We also added a short argument for using the median parameter set and extended the sentence to make it more clear what the median parameter set is. Here some more detailed information for clarification: The 100 calibrations done with each objective function result in 100 optimized parameter sets. These hundred different parameter sets lead to very similar model performance in calibration, which is a common observation usually referred to as *equifinality* (Beven and Freer, 2001). The equifinality concept rejects the idea of one single best parameter set. We calculated the median model performance of all 100 parameter sets in calibration and validation to not select the best, but rather a representative value from the efficiency distribution.
**Modification:** P7 L1-2

**P6 L10:** Should be in the chp where SFC are described.
**Reply:** We moved the sentence to chapter 2.2.
**Modification:** Sentence moved from P7 L3 to P5 L1-2

**P6 L22:** Did you not optimize on both these criterions...if not that must be mead clear earlier. If you did...then it should result in 26 parametersets...
**Reply:** Correct, it's two times 13 resulting in totally 26. We adapted the sentence to make that clear.
**Modification:** P7 L18

**P6 L22:** I assume the calibration results in a optimal parameterset given the criterion used. And not in a simulation...
**Reply:** A calibration results in both an optimal parameter set and its corresponding runoff simulation.

**P6 L24:** But what about variability? Is it not better to have high variability and some good model simulations than low variability around poor simulations (as I assume a low score indicates)
**P6 L24:** here you say you uses both Isingle and the combined I signle and I reff ...my previous comment asks about this...
**Reply:** The two comments above relate to each other and thus we answer them together:
The optimal value for the normalized SFC error is 0 (see Table 3). In the text we describe the variability of the error value and its magnitude for the objective functions $R_{eff}$, $I_{Single}$ and $I_{Single\_Reff}$. The best situation would be a low variability at a low error magnitude. Low variability at a high error magnitude would indicate that the related objective function is not suitable for model calibration aiming at the respective SFC. High error variability combined with some low error magnitudes would indicate that certain SFCs used as objective function ($I_{Single}$) can lead to good estimates. But it also means that not all SFCs used in $I_{Single}$ result in good SFC estimates. This fact supports our main conclusion that SFCs should preferably be estimated from targeted runoff model calibration.

**P6 L27:** In general or for TA1?
**Reply:** The description refers to the results of TA1 which was selected as a representative example. But overall a similar pattern can be seen for all SFCs. We rearranged this paragraph (also because Fig. 4 in the unrevised manuscript was removed) and it should be more clear now.
**Modification:** Sentence was moved within paragraph from P7 L29 – L31 to P7 L19-21

**P6 L28:** illustrated by...
**Reply:** Was added.
**Modification:** P7 L27

**P7 L1:** Again...what is median simulation?
**Reply:** For each SFC we get each 13 normalized errors for the objective functions $I_{Single}$ and $I_{Single\_Reff}$ (as shown in Fig. 3). With "median" we referred to the median of these 13 normalized SFC errors from calibrations with $I_{Single}$ and $I_{Single\_Reff}$. The sentence of P7 L1 was deleted due to restructuring of the paragraph.
**Modification:** P7 L28-29

**P7 L1:** Reff of <0.1 is very poor and telle me that there are something substatially wrong. Do I misunderstand? Reff should be higher than at least 0.5 before you can say you have representative data and higher than at least 0.7 before you can say that you are able to model a catchment satisfactory.
**Reply:** The Nash-Sutcliffe efficiency was not displayed in Fig. 4 (here we refer the Fig.4 in the unrevised manuscript). For more discussion on the Nash-Sutcliffe efficiency please see comment 1 in RC 1.

**P7 L6:** The Reff is very low for all catchments it seems like. only one above 0.5. This tells me the opposite of you comment here! Why is Reff so low. Poor precip data og runoff data or??? Is a acheived criterion below 0.2 good agreement with obeserved? In that case you have to explain how.
**Reply:** We added some text in the results part of the manuscript and also adapted some figures to clarify the confusion regarding the Nash-Sutcliffe efficiency and to avoid any misunderstanding (see comment 2 of RC1). In the original version of Fig. 4 (corresponds to Fig. 5 in the unrevised

manuscript), the y-axis of $R_{eff}$ (right axis) ranged from 0 on top to 1 at the bottom. The closer to 1 (thus to the bottom), the better is the Nash-Sutcliffe efficiency. For all 5 types of objective functions the value of Nash-Sutcliffe was above 0.5. Figure 4 was changed and now shows the difference in model performance between a model calibration with $R_{eff}$ and model calibrations with $I_{Single}$, $I_{Single\_Reff}$, $I_{Multi}$ or $I_{Multi\_Reff}$ (see comment 2 in general comments RC1).

**Review 2: comments, response and modifications**

Dear Björn Guse,

Thank you for your efforts with our manuscript. We greatly appreciated the comments, which helped us to improve the manuscript. Please see below our detailed response to each of the comments.

Best regards,

Sandra Pool and Co-Authors

**Major comments RC 3**

**Comment 1:** In its current state, the article is in my opinion mainly related to hydrology (also by the selection of the journal). At several parts the authors emphasized its ecological importance (e.g. Page 1, Line 10, P.2, L.1-2, P.11, L.14-16). However, I do not really see a connection to ecology. Thus, either the article has to be fully focused on hydrology or it is required to emphasize its ecological relevance

**Reply 1:** We agree that our modelling approach for estimating SFCs is a typical hydrological approach. However, the motivation for our study is strongly related to ecohydrology because of its focus on the simulation of ecologically relevant SFCs of our study catchments (chapter 2.2). While many of our SFCs are also the focus of other ecohydrological studies, they are untypical for purely hydrological modelling studies where signatures such as segments of the FDC or runoff ratio are of interest. Given the importance of SFCs in ecology we find it important that the hydrological community addresses the issue how to compute these values for ecological studies and applications.

**Comment 2:** In this article, the Nash-Sutcliffe Efficiency is used as an example for a traditional calibration approach. However, in recent studies the use of several performance measures related to different parts of the hydrological system is recommended. In particular the use of typical performance metrics such as Reff or Kling-Gupta-Efficiency (KGE) in combination with signature measures is recommened (see eg. Van Werkhoven et al., 2009). Thus, I think that a comparison only with the Reff is not sufficient. I recommend to use two or three performance measures such as PBIAS or KGE to show that SFCs also outperforms a calibration approach based on NSE in combination with PBIAS (or other performance measures).

**Reply 2:** We agree that combined objective functions based on e.g. Nash-Sutcliffe efficiency ($R_{eff}$) and volume error are widely used for runoff model calibration. We actually used such combined objective functions (that are not based on SFCs) in our previous study (Vis et al., 2015) to estimate the same SFCs of the same catchments as in the current study. The average SFCs error (percent error) in calibration was lowest for calibrations with $R_{eff}$ (error of -4.9 %), followed by calibrations with objective functions based on a) $R_{eff}$, LogReff and volume error (error of -5.1 %), b) $R_{eff}$ and volume error (error of -5.6 %) and c) $R_{eff}$, MARE, spearman rank correlation and volume error (error of -6.1 %). Some other combined objective functions were also tested, but resulted in clearly poorer SFC estimates than the ones listed above. We therefore decided to use $R_{eff}$ as a benchmark in the current study. We added some information about the results of the preceding study in the introduction of our manuscript.
**Modification:** P3 L2-6 and P3 L25-28

**Comment 3:** Moreover, it needs at least to be discussed how a calibration approach based on these SFCs is related to recent studies using hydrological signatures such as segments of the FDC (see Yilmaz et al., 2008, Pfannerstill et al., 2014).

**Reply 3:** Thank you for making us aware of the study from Yilmaz et al. (2008) which we mentioned in the introduction in addition to the study of Pfannerstill et al. (2014). We also used the two studies in the first part of the discussion to briefly discuss our results.

**Modification:** P2 L25-26 and P3 L16-19 (introduction), P10 L33-P11 L1-3 (discussion)

**Comment 4:** P. 1, L. 24-27: I do not fully agree with this statement. There are different ways of how to calibrate a discharge time series. Certainly there are studies which are focused on certain parts such as on high or low flows. However, other studies aim to represent the whole hydrological system at best without neglecting or emphasizing certain parts. In the way towards a good representation of the hydrological system, the latter one should be the general goal and a strong focus on certain parts of the hydrological system should be a specific case.

**Reply 4:** We agree that some general agreement often is the calibration goal. Our statements aims at the fact that even with such a general agreement, specific aspects might be poorly simulated as we have shown in the preceding study (see Vis et al., 2015). We adapted the sentence to include your input.

**Modification:** P2 L1

**Comment 5:** P.2, L. 21: Here, SFCs are defined "as equivalent to hydrological signatures". In this case, I do not understand the use of SFC. It is stated before that hydrological signatures are a more common term in hydrology. It is certainly required to justify why you used the term SFC since I do not really see the strong relationship to ecology.

**Reply 5:** We added some information about the reason for using the term SFC in our study in the introduction. As described in the reply of comment 1, the selection of the SFCs is motivated by their ecological relevance in the study catchments. While both terms, hydrological signature and SFC, describe characteristics of the hydrograph, only the term SFCs makes a distinct connection to ecology. The term SFC has also been used for many years, whereas the term hydrological signature has been introduced more recently.

**Modification:** P2 L 29-30

**Comment 6:** I think that the article would benefit from a more detailed interpretation of the results. For example on P. 7. L. 12-15; P.8, L.17-18: Can you explain these results or more specifically the behaviour of these SFCs?

**Reply 6:** The behavior of SFCs regarding robustness (comment on P8 L17-18) could be explained by grouping them into SFCs that represent catchment characteristics, average flow conditions or dependency on inter-annual weather changes. The striking behavior of the SFCs TL1 and MH10 might be related to the fact that they are calculated on one single value. These two SFCs as well as the pattern in the robustness are discussed in the second and third paragraph of chapter 4.1. We couldn't find a consistently different behavior for the remaining SFCs and therefore it was difficult to find reasonable explanation for their behavior (e.g. why is FL2 estimated similarly well with $I_{Sinlge}$ and $R_{eff}$, but worse with $I_{Sinlge\_Reff}$?) (comment on P7 L12-15). We adapted the text of P7 L12-14 to include the results about MH10 and removed the statement about FL2 and MA26. This should guide the reader's attention to the more exceptionally behaving SFCs.

**Modification:** P8 L14-17

**Comment 7:** It could be interesting to analyze the relationship/correlation between the SFCs.

**Reply 7:** We did a Spearman rank correlation test for the 13 SFCs using all 25 catchments (Table 1 of this response). We did the correlation test for both modelling time periods. The correlation values of the two time periods were similar and therefore averaged.

**Comment 8:** The combination of different metrics might outperform in general single-metric approaches. However, the more metrics are included, the more a trade-off might occur and the equifinality problem arises. In this context, can you give a recommendation for a good number of required SFCs? In the best case, a systematic way of how to select the best SFCs will be provided. Even though when I expect that it is difficult to find a precise number, it is worth discussing this point.

**Reply 8:** Adding more criteria into a combined objective function usually rather decreases than increases the equifinality issue according to our experience and other studies, although we agree with you that also a trade-off between the different criteria might occur.

We systematically selected SFCs for a multi-objective function based on the criteria of information value and robustness. Our results indicate the importance of jointly evaluating both criteria for the selection of SFCs for a multi-objective function. However, we cannot give a minimum number of SFCs required for such an objective function, because this will depend on the type and combination of SFCs one is interested in. We could show that the four SFCs used for the multi-objective function preserved similar hydrograph characteristics as the Nash-Sutcliffe efficiency (similar estimation accuracy of SFCs not included in the objective function; Fig. 4b and 7c). How much value additional SFCs have for the representation of the hydrograph characteristics would have to be evaluated using many more than 13 SFCs and eventually using synthetic data to also see the effect of redundant information (see discussion chapter 4.3). We made some small changes to chapter 4.3 to take some of the suggestions and questions of comment 8 into account.

**Modification:** P12 L23-25 and P12 L28-32

**Comment 9:** The figures 3-8 are very similar (at least visually). I think that the article would benefit from emphasizing the relevance of each figure. It is partly difficult to differentiate them. Maybe you can also thinking about reducing the number of figures to improve the overall message. For example, the figures 6 and 8 have almost the same figure caption. To summarize this point, it is easier to detect the whole message in the case of a more distinct presentation of the results. One example for this is the figure 9 which can be clearly distinguished from the other figures. These results are easier to understand.

**Reply 9:** We agree that some figures are similar and maybe hard to distinguish. We decided to change two figures and delete one to present the results in a more interesting or intuitive way in the revised manuscript: Figure 3 was changed to a 3-D graph which on one hand allowed us to show all the relevant information in one single plot and on the other hand reduced the number of subplots from 6 to 2. We removed Fig. 4 of the unrevised manuscript, because its main information/ conclusion is similar to the information of Fig. 3 and some of its information is contained in Fig.6b. Figure 4 was adapted so that it shows model performance as the difference between a model calibration with $R_{eff}$ and model calibrations with $I_{Single}$, $I_{Single\_Reff}$, $I_{Multi}$ or $I_{Multi\_Reff}$. This change helps to directly compare SFC-based calibration approaches with a traditional Nash-Sutcliffe calibration approach and therefore provides a more clear answer to question 3 in the introduction. We also adapted the axis labels of Fig.s 6, 8 and 10 (corresponds to Fig. 5, 7 and 9 in the revised manuscript).

**Modification:** P25-26 (Figure 3), P27 (Figure 4 of the unrevised manuscript), P28-29 (Figure 4), P29-30 (Figure 5), P31 (Figure 7), P33 (Figure 9)

**Comment 10:** P. 8, L.12-13: Could you specify how you can here differentiate between error dependence on time period or objective function?

**Reply 10:** We adapted the sentence. Generally it means that we looked for systematic patterns in the error magnitude (absolute value of normalized SFC error) of the SFCs. For some SFCs (e.g. TL1) the error magnitude could be considerably higher in one modelling time period than in the other one. We described the error of such SFCs as being time depended. Some other SFCs had clearly higher error magnitudes when calibrated on a certain objective function (e.g. MH10). The estimation accuracy of such SFCs was considered as being dependent on the objective function.

**Modification:** P9 L19-21

**Comment 11:** P.8, L.14-15: I do not understand this statement that the SFCs are neither related to flow components nor to flow conditions. Hydrological signatures (as an equivalent term) are known to be of special importance to explain the hydrological behaviour. Thus, what can we learn from using these SFCs in terms of the hydrological behavior in the catchment. And how is this related to the general idea of the hydrological signatures?

**Reply 11:** We are sorry for the confusion this statement evoked and made the sentence clearer. We meant that we could not relate the estimation accuracy (error magnitude, spread of error magnitude and dependency of error magnitude on modelling time period/ objective function (see comment 10)) of the analyzed SFCs to the flow components or flow conditions they belong to. E.g. we cannot say that all high-flow related SFCs had very low estimation accuracies.

**Modification:** P9 L21-22

**Comment 12:** P.8, L.26 to P. 9, L.21: I agree with this part which is clearly understandable, but certainly also not surprising. It is mostly existing knowledge of hydrological modelling. What can we learn here except of using several and different metrics. I recommend to shorten this part and emphasize the most important points from this study. In contrast, I really like to following passage (P. 9, L.21-31).

Please also discuss the impact of a SFC-based calibration for the process representation. Can you state that the hydrological system is overall better represented by using several SFCs?

Please discuss the benefit of optimizing one specific SFC. This leads to a modelled hydrograph which is able to represent a very specific condition but probably not the overall hydrograph. This implies that the part of the hydrological system which is not in the focus of this SFC is probably not adequately considered. This might be of particular relevance when using very specific SFCs such as MA26.

**Reply 12:** Thank you for this helpful comment. We adapted the discussion part you mentioned focusing on your proposed aspect of how the calibration with SFCs affects process representations: The benefit of optimizing one specific SFCs lies in the relatively accurate estimation of the respective SFC compared to a calibration with $R_{eff}$ or a multi-SFC objective function. Model calibration on one single SFC clearly emphasizes the hydrograph aspects of the selected SFC which can negatively affect other hydrograph aspects. This implies that calibrations with $I_{Single}$ can lead to poor model performance for SFCs not included in the objective function (Fig. 3). E.g. calibrating on the frequency of moderate floods (FH6) leads to poor model efficiencies for base flow (ML20) (Figure 6b) which indicates that the representation of the main runoff processes can suffer by SFC-specific model calibration. From the fact that a calibration with $R_{eff}$ and a calibration with multiple SFCs lead to comparable SFC estimates we infer that the main hydrological processes of the catchments are similarly well represented with the two approaches. We assume that these two calibration criteria result in a better process representation than the calibration with a single SFC, because they outperform the calibration with $I_{Single}$ for those SFCs not included in $I_{Single}$.

**Modification:** P10 L6-33

**Minor comments R 3**

**P.1, L. 12:** maybe "optimization" instead of "minimization or maximization".
**Reply:** We did the replacement as suggested.
**Modification:** P1 L13

**P.1., L.16:** Are these over- and underestimations a general aspect of these SFCs or are they case-specific?
**Reply:** These results represent the general tendency of the model and the results from a specific objective function and/or modelling time period can deviate from these general conclusions (see Fig. 8 in the revised manuscript).

**P.2, L.16-19 and L. 30:** I recommend to include the study from Yilmaz et al. (2008).
**Reply:** Thank you for this suggestion. We included the study of Yilmaz et al. (2008) in the introduction.
**Modification:** P2 L25-26 and P3 L16-20

**P.2, L. 34:** The meaning of "esoteric and subtle aspects of the flow regime" is unclear.
**Reply:** Good point, we adapted the sentence.
**Modification:** P3 L15

**P.3, L. 19:** Please think about renaming the section to "Methods and materials", since a catchment is not a method.
**Reply:** We changed the title from "Methods" to "Materials and methods".
**Modification:** P4 L8

**P.5, L.31:** Why you have used 0.2 and 0.25 as weights?
**Reply:** $I_{Multi}$ consists of four SFCs that we wanted to weight equally, which results in a weight of 0.25 for each of the SFCs. These four SFCs were combined with the Nash-Sutcliffe efficiency ($I_{Multi\_Reff}$) and each of the components was again assigned the same weight (0.2). We weighted all components of $I_{Multi}$ and $I_{Multi\_Reff}$ equally because there was no clear reason to weight one/some of the SFCs differently from the others.

**P.6, L. 9:** Could you specify "median parameter set"?
**Reply:** This sentence should say "…, the median model efficiency of each catchment was selected." We changed this sentence.
**Modification:** P7 L1

**P. 11, L.15:** Could you specify "later application of simulated SFCs related to flow alteration – ecosystem change relationships". This aspect was up to now neither in the focus of the article nor emphasized as an overall aim.
**Reply:** This statement refers to the first paragraph of the introduction where we motivate the need for accurate SFC estimates by giving the example of flow alteration – ecosystem change relationships used for sustainable flow management. We adapted the sentences related to the mentioned statement to make this connection more clear.
**Modification:** P13 L13-16

**Table 1:** TA1: runoff with two f
**Reply:** We added the missing f.
**Modification:** P19

**Table 1:** Why you have named the SFCs FH6 and FH7 and not FH3 and FH7.
**Reply:** We agree that the abbreviation can be confusing, but decided to follow the abbreviations used in previous publications and in the EflowStats R-package that was used for the calculation of the SFCs. The same abbreviations are commonly used in many studies (see e.g. Olden and Poff, 2003).

**Fig. 5:** Can you explain the outlier TL1?
**Reply:** We discussed the low estimation accuracy of TL1 in the discussion (last paragraph of chapter 4.1).

**References used in the responses to the reviewer**

Beven, K., and Freer, J.: Equifinality, data assimilation, and uncertainty estimation in mechanistic modelling of complex environmental systems using the GLUE methodology, J. Hydrol. 249, 11–29, doi:10.1016/S0022-1694(01)00421-8, 2001.

Olden, J. D., & Poff, N. L.: Redundancy and the choice of hydrologic indices for characterizing streamflow regimes, River Research and Applications, 19(2), 101-121, doi: 10.1002/rra.700, 2003.

Vis, M., Knight, R., Pool, S., Wolfe, W., and Seibert, J.: Model calibration criteria for estimating ecological flow characteristics, Water, 7, 2358–2381, doi:10.3390/w7052358, 2015.

*Table 1: Spearman rank correlation coefficient between the 13 ecologically relevant streamflow characteristics used in this study.*

[revised manuscript text omitted]

---

## Referee Report (RR1)

[referee-annotated manuscript omitted]

---

## Referee Report (RR2)

Review of manuscript no.: hess-2016-546 by Sandra Pool et al: Streamflow characteristics from modelled runoff time series - importance of calibration criteria selection

This paper analyses the informational value of different streamflow characteristics (SFCs) used in calibration of a hydrological model, as alternative and/or supplement to traditional calibration criteria like Nash-Sutcliffe efficiency criterion etc. The motivating application is estimation of ecologically relevant SFCs by precipitation-runoff modelling, ultimately in ungauged catchments. The current paper, however, does not address the regionalisation aspects of this challenge (as is clearly stated on page 3, line 28). It is nevertheless an interesting paper bringing together traditions within classical hydrological model calibration and eco-hydrology.

The paper is well written and easy to perceive. Still, I do have some concerns about some aspects of the manuscript, in particular as focus shifts from calibration evaluation to estimation of SFCs. I will elaborate on this below.

The paper contains no clear definition or distinction between a goodness-of fit measure based on an SFC, and a 'traditional calibration criterion' (TCC). In my view, a comparison of SFCs to TCCs should acknowledge that there may be a transition zone between the two, but still provide a clear distinction which makes the comparison meaningful. SFCs are (citing): 'often used to refer to <such> specific aspects of the flow regime', however, this is also true for $R_{eff}$, $R_{eff}(ln)$, slope of FDC etc. In addition, the analysis use $R_{eff}$ alone as TCC benchmark, although the necessity of combining different TCCs in calibration is well recognised. This constructs a comparison in favour of SFCs.

A definition of 'traditional criterion' could be used, for instance along the lines of 'A goodness of fit measure computed from all sim-obs residuals within the calibration period, possibly transformed'. Then distinct groups of SFCs could be recognised by being 'selected from specific seasons or situations', or 'based on duration of events or conditions' etc.

With no rules for how SFCs are constructed, statements like 'High flow SFCs tend to be under-estimated' are meaningless. Flashiness index and concentration time both characterise high-flow behaviour, but underestimating one would mean overestimating the other. From table 1 the reader may verify that each SFC used in this paper is scaled so its value is positively related to flow magnitude for the relevant section of the FDC, but this should be asserted in the text when used in conclusions.

Most serious objection: The paper draws conclusions based on subjective evaluations of results. There is no reference to statistical significance or confidence intervals, and very few references to thresholds for 'good simulation', 'small error' etc. In section 2.4.1 on page 5, it is stated that an ensemble of 100 calibrated parameter sets were available for each objective function, allowing analysis of parameter uncertainty. I have not investigated the genetic algorithm used for calibration, but could the variability within this ensemble be used to assess which differences extend beyond mere noise?

Details and specifics:

Page 3, lines 22-24. This sentence is unclear and possibly erroneous. What is meant?

Page 3 line 29 and onwards. Please specify also what is **not** used. For instance in (1) make it clear that this SFC is used alone, in (2) whether or not the multi-SFC vector includes the SFC being evaluated, and (3) if the TCCs are kept in or excluded when the SFCs are included.

Page 4 line 13. The reference to NAVD 88 is unnecessary.

Page 6 line 10: Not yet knowing that the SFCs are normalised to a common scale, the reader may wonder how $R_{eff}$ and an arbitrary SFC can be equally weighted. Just put in a 'normalised (see below)'.

Page 6 line 20: Can you specify how small error are required for a SFC to be robust, and how 'relatively good simulations for other SFC' are required for being informative? These limits have the impact of restricting which SFCs enter the Multi alternative.

Sections 3.1.1 and 3.1.2: Would these come more naturally in opposite order? I perceive the $I_{single}$ experiment as the simplest and most obvious, the one-to-any SFC as more involved.

Would it be informative to summarise in a table for all SFCs what is illustrated in fig 3 for TA1?

Page 8 line 6. What is required to deserve a 'well simulated' mark? Is there an a priori defined threshold?

Page 8 line 8. Se discussion above about SFC estimates being high or low.

Section 3.3, page 5: This is one of the weaker parts. See the above point on subjective conclusions.

The categorisation in lines 19 and 20 are well defined, but then not used for anything. The lowest error class collects everything from perfect match to 10% error, capturing the result for all the mean-flow SFCs, 3 out of 4 low-flow SFCs and 4 out of 5 high-flow SFCs. Still, this paragraph states that all high-flow SFCs and three of four mid-flow SFCs are under-estimated, whereas all except one low-flow SFC were over-estimated' (lines 21-22). Such conclusions need to refer to at least a clearly stated threshold, but preferably to statistical significance.

The SFCs listed as having small vs medium absolute errors in line 23 does not correspond to a sorted grouping of the errors in fig 8. The FH6 error is characterised as 'small', but is larger than the DH16, MA41 and TA1 errors listed as medium, as well as the MA26 and FL2 errors not listed in these two lowest groups.  Likewise it is difficult to see from figure 8 why MA26, DH13 and FH7 are identified in line 25 as having large error ranges, while MA41, TA1 or DH16 are not. One gets to suspect that the text is referring to another version of figure 8.

The statement in lines 15-17 suggests an identification between goodness of fit and process representation, which this paper neither investigates nor justifies.

Figure 9: The term 'absolute normalised SFC errors' are used. The figure seems to display signed errors.

Any calibration criterion as used in this experiment is a compression of the entire vector of simulation residuals into a scalar goodness-of-fit (GOF) measure. An SFC can provide a narrow-band, highly specialised GOF, whereas the traditional TCCs are 'broadband' GOFs aiming to minimise the expected error in any situation. I question the practical relevance of $I_{single}$ calibration, but the idea that a general, multi-purpose GOF can be constructed from combining several specialised SFCs, in my opinion deserves investigation.  With the mentioned weaknesses improved, this paper is a valuable piece in that puzzle. A continuation along this path should investigate possible conflicts between SFCs, and elaborate more thoroughly on uncertainty and identifiability.

---

## Referee Report (RR3)

Second review of manuscript HESS 2016 – 546

Sandra Pool, Marc J. P. Vis, Rodney R. Knight and Jan Seibert

Streamflow characteristics from modelled runoff time series – importance of calibration studies

The second submitted version of this manuscript have improved compared to the first, and I can now recommend its publication in HESS.

The original review had three major and 12 minor comments. The two first major comments have mainly been met by explaining in more detail how the authors define and use 'streamflow characteristics' (SFCs) as opposed to traditional performance measures (PMs) or objective functions. The lack of precise definitions remains, but the changes made do make it easier for the reader to perceive what is meant.

The last major comment has been met by carrying out one additional analysis, with a more rigorous approach to statistical significance. This is a valuable extension and substantial improvement of the paper. Having said this, some subjectivity still in my view weakens the analysis and discussion, but not to the extent that the conclusions drawn are not justified.

The list of specific minor comments have been properly addressed, in most cases by making the suggested adjustments. A suggestion in comment #7 has not been followed, which is all OK. Likewise no change is made with respect to comment #11, but the corresponding passage in the discussion expresses more an opinion than a claimed result, and is also accepted.

Sjur Kolberg

---

## Author Response (AR2)

**2nd revision of manuscript**

Journal:         Hydrology and Earth System Sciences (HESS)
Title:           Streamflow characteristics from modelled runoff time series - importance of calibration criteria selection
Manuscript:      hess-2016-546
Authors:         Sandra Pool, Marc J. P. Vis, Rodney R. Knight, and Jan Seibert

Dear editor,
We thank you for your and the reviewer's efforts with our manuscript. The two reviewers provided again valuable comments on our manuscript, which helped us to further improve the clarity of our manuscript. Below, we reply to each of the comments from the reviewers and indicate the changes that have been done accordingly (marked with blue color). References to pages (P), lines (L), chapters, figures and tables refer to the track-changed revised version of our manuscript. We also did a few additional minor edits that are marked in the track-changed manuscript.

Yours sincerely,
On the behalf of all Co-Authors,
Sandra Pool

**Reviewer 1: comments to 1st revision, response and modifications**

**Comment 1:** The ecological relevance would be relevant to say something about in relation to or within table 1. In table 1 many things is repeated i in name and definition. A description of relevance is more informing... not crucial but recommended for a better reader experience

**Reply 1:** We agree that column one (name) and three (definition) of Table 1 contain redundant information for some SFCs, especially for the SFC MA41. For most other SFCs the name column provides a short and concise description of the SFCs, whereas the definition gives additional information on the calculation of the SFCs. We therefore would like to keep these two columns separated. To emphasize the difference between the two columns we renamed the column "definition" to "further explanation".
The direct and explicit ecological relevance of SFCs for the biodiversity is indeed interesting. Nevertheless, we would like to refer to e.g. the study of Knight et al. (2008) for a description of the specific influence of the selected SFCs on the fish diversity in our study catchment.
**Modification:** P18-19 Table 1

**Comment 2:** Clearify the difference between 1 and 2....is one when one and the same SFC is a part of the objective funtions and 2 when many and other SFC is used in the objective functions
**Reply 2:** We specified the sentence by writing "… contains one or multiple other SFCs?"
**Modification:** P4 L3

**Comment 3:** As commented....name and definition gives in most cases double information.
**Reply 3:** Please see reply on comment 1.

**Comment 4:** This is wrong ...need fixing
I multi formula with 0.25 (I1+...+In) and n=13 I eff can not be max 1. Only if n=4 it can be max 1
**Reply 4:** We apologize for the inexact formulation of this sentence and the resulting confusion. The objective function $I_{Multi}$ consisted of 4 SFCs, each of them weighted 0.25. However we agree that at this point in the text, the reader does not know how many SFCs actually will be included in $I_{Multi}$, because the number of SFCs used for $I_{Multi}$ is part of the study results. We therefore adapted the

sentence and also the corresponding Table 2 and replaced the weights by a formulation using n as a variable.
**Modification:** P6 L15 and 17 (text) and P20 (Table 2)

**Comment 5:** So what you say here is that when calibrating on one SFC the other SFC's became poorly simulated. But in relation to what? To calibrating only on Reff or?
**Reply 5:** We adapted the sentence to relate the model efficiency with $I_{\text{Single}}$ to calibrations with $I_{\text{Single\_Reff}}$ and $R_{\text{eff}}$.
**Modification:** P7 L16

**Comment 6:** Is that Isingle_Reff you mean...in that case write this to avoid confusement... ...is that Isingle_Reff you mean?
**Reply 6:** Yes, we meant $I_{\text{Single}}$. However, this sentence was deleted due to restructuring this paragraph.
**Modification:** P7 L16-17

**Comment 7:** Figure 3 is very difficult to get any information from. This was a very confusing figure not illustrating very well the point. Need a fix... Either you need a more understandable figure where it is also possible to separate the different dots and start etc or you take away this plot.
What is the different colors showing etc... a legend is at lest needed
**Reply 7:** We adapted Fig. 3 by reducing it from 3-D to 2-D, added a legend and adapted the figure caption.
**Modification:** P24, Fig. 3

**Comment 8:** This is an very important statement here i think. Where is that illustrated? It seems unrealistic when Reff is down to 0,68... You need to documented this better than saying almost 100%...
**Reply 8:** We had a plot showing the almost perfect model performance for each SFC during calibration in our first version of the submitted manuscript. However, that figure was removed and replaced with Fig. 4 that shows the calibration performance with $I_{\text{Single}}$ in relation to the model efficiency with $R_{\text{eff}}$. Instead of adding the previous plot again, we now inserted the exact numbers of the absolute normalized SFCs in calibration in the text.
**Modification:** P7 L26-27

**Comment 9:** Could this be shown clearly in a figure. This is the most sentral finding here as I read this.
**Reply 9:** This finding is shown in Fig. 4a and b. We added the reference to this figure in the text.
**Modification:** P7 L29 and 30

**Comment 10:** Do you mean that you calibrate with a combined Isingle_Reff and Reff than Reff is double included? or do you mean that you use Isingle and Reff as described in Isingle_Reff?
**Reply 10:** We refer to calibrations with either $I_{\text{Single\_Reff}}$ or $R_{\text{eff}}$. We adapted the text accordingly.
**Modification:** P7 L30

**Comment 11:** I can under stand from 4 a and b how Reff changed in average for different SFC tests (this is well described). But what values do the SFC dots refer to? What difference in model performance do they represent. Is the axis title right for tis dots?
It is also very difficult to read this figure and distinguish the different perf measures... Need improvement
I understand the dots and different coulors in I singel, but in I multi several of the SFC is included in the criteria and then I do no understand the dots if they ar not certain combinations...this need better decription ...
**Reply 11:** Figure 4a and b show the model performance for calibrations with $I_{\text{Single}}$, $I_{\text{Single\_Reff}}$, $I_{\text{Multi}}$ and $I_{\text{Multi\_Reff}}$ relative to the model performance for calibrations with $R_{\text{eff}}$ (difference in model performance is calculated). The black rectangles are the model performance in terms of $R_{\text{eff}}$ and all white rectangles

are the model performance in terms of MARE. All colored circles represent model performance for one of the thirteen SFCs (nSFC, see Table 3). We made some adaptations to the legend, axis label and figure caption.
**Modification:** P25 Figure 4

**Comment 12:** There are two outliers in the group...they are no commented why?
**Reply 12:** We assume the reviewer refers to the SFCs MH10 and TL1, because the behavior of these two SFCs is striking. We mentioned the outlier MH10 in the results part (P7 L28-29) and discussed it in the last paragraph of chapter 4.1. We didn't mention TL1 at this point of the results, because TL1 is not an outlier regarding the performance related to a certain objective function. However, TL1 is an outlier in terms of magnitude. This is mentioned in chapter 3.3 of the results and discussed in the last paragraph of chapter 4.1.

**Comment 13:** What do you try to say here? Unclear to me... What is the point of this statement? And figure 6b ...about the Open cricles ...what extra do they tell then waht is told in 6a...and the multi comment.. does that say that this is the value for that single SFC when using Multi as criterion? Thsi must be clearer stated.
**Reply 13:** The statement refers to Fig. 6b which shows the information value of the four SFCs selected for $I_{Multi}$ for each of the 13 SFCs. We adapted the paragraph to make that more clear. We also made some adaptations to Fig. 6, namely to the legend, figure caption and title.
**Modification:** P8 L9-16, P27 Figure 6b

**Comment 14:** This is important and need to be illustrated clearly.
**Reply 14:** This finding is illustrated in Fig. 4a as mentioned in the text (P8 L20).

**Comment 15:** Is that so strange that it need saying... Reff was poorer when calibrating with other criterias than Reff than calibrating with Reff alone... the opposite should be impossible...
**Reply 15:** We agree that this statement describes a result that can be expected. We therefore removed the statement.
**Modification:** P8 L19-20

**Comment 16:** This is also something that is too logic and should be so obviously expected that it does not need mentioning...
**Reply 16:** We agree that this statement describes a result that can be expected. We therefore removed the statement.
**Modification:** P8 L23-25

**Comment 17:** also obvious and draw attetntion away from more important findings.
**Reply 17:** We agree that this statement describes a result that can be expected. We therefore removed the statement.
**Modification:** P8 L25-26

**Comment 18:** Do you then mean SFC found in model calibrated by Reff...need fixing to be clear.
**Reply 18:** We meant the objective function $R_{eff}$. We added "…the objective function $R_{eff}$" to the sentence to make it clear.
**Modification:** P8 L29

**Comment 19:** This is something too obvious again...
**Reply 19:** We agree that this statement describes a result that can be expected. We therefore removed the statement.
**Modification:** P9 L30 – P10 L2

**Comment 20:** ..is this explicitly shown in the figures. I can not see that commented earlier.

**Reply 20:** Yes, this is explicitly stated at P8 L27-30 and shown in Fig. 4b and 7c.

**Comment 21:** ..this should again be obvious when using Imulti...but not so when using Reff..so mixing these in the same statement is confusing

**Reply 21:** We removed this sentence.

**Modification:** P10 L4-6

**Comment 22:** is this so? I assume you calibrate you model and use you model for a purpose and calibrate it for that and not all other SFC that for your case is not relevant. The mor interesting question is the robustness of a model that do not capture all SFC...

**Reply 22:** We agree that our results clearly indicate that the model should be calibrated on the SFC of interest. However, one might be interested in many different SFCs and it would therefore be practical if a single model calibration resulted in a simulated hydrograph from which all SFCs could be accurately calculated. We discussed this aspect in chapter 4.3 of the discussion. We adapted the mentioned sentence to make that more clear.

**Modification:** P10 L8

**Comment 23:** Can this have something to do with the model structure of HBV and the over parameteization in the model that makes it possible to calibrate very well but are not robust as it is not physically based and thus only works on situations it is calibrated against..

**Reply 23:** Yes, we agree that the model structure could be a further explanation for the poor robustness of a SFCs in some catchments. There is a short statement on that on P10 L27-28. Using a physically based model instead of a conceptual runoff model does not necessarily improve the results (see discussion chapter 4.5) because they also rely on calibration.

**Comment 24:** from recent year and a 13 year old reference?

**Reply 24:** We removed the term "from recent years".

**Modification:** P12 L10

**Comment 25:** Is it not also a question wheter a model calibrated over some year is in average good but not really good on spesific SFC that occure more periodically? a model calibrated for periodes including winter conditions is that likely to equally good on dry summer periods? is it an idea to calibrate models more for defined caracteristic periods and use different models depending on the situation you are in... than on in average good model?

**Reply 25:** We agree and also have shown with our results that a model with a good average performance (e.g. measured with $R_{\text{eff}}$) over several years does not necessarily perform well on specific SFCs occurring more periodically. Also, a model often performs best on conditions it was calibrated on. As you suggested, model calibration could not only be focused on SFCs, but also on the periods most relevant for a certain SFC. SFCs that are subject to inter-annual weather changes (e.g. FH7 – frequency of larger floods) could probably benefit most from such a calibration approach. Caldwell et al. (2015) therefore conclude that the calibration process probably has as much influence on SFC estimates as the model structure (P13 L1-2).

**Comment 26:** this can not be correct with n>4

**Reply 26:** We adapted the equation in Table 2 (please see reply on comment 4)

**Modification:** P20 (Table 2)

**Reviewer 2: comments to 1[st] revision, response and modification**

**Comment 1:** I thank the authors for carefully revising the manuscript. I am very satisfied with the answers to my comments. Thus, I suggest to accept the manuscript as is. I have only one spelling comment: P.9, L.31: surprising instead of surprizing.

**Reply 1:** Thank you for this comment. We corrected the spelling mistake.

**Modification:** P10 L8

[revised manuscript text omitted]
_{eff}$ | $1 - \dfrac{\sum (Q_{obs} - Q_{sim})^2}{\sum (Q_{obs} - \overline{Q_{obs}})^2}$ | 1 |
| Efficiency for each individual SFC[1] | $I_{Single}$ | $1 - \dfrac{\lvert I_{obs} - I_{sim} \rvert}{I_{obs}}$ | 1 |
| SFC and model efficiency | $I_{Single\_Reff}$ | $0.5\,(I_{Single} + R_{eff})$ | 1 |
| Efficiency for the selected SFCs[2] | $I_{Multi}$ | $\dfrac{1}{n}\,$$(I_{Single_1} + \ldots + I_{Single\_n})$ | 1 |
| SFCs and model efficiency | $I_{Multi\_Reff}$ | $\dfrac{n-1}{n}\,$$I_{Multi} + \dfrac{1}{n}\,$$R_{eff}$ | 1 |

[1]For each of the 13 SFCs a specific $I_{Single}$ exists.

[2]$I_{Multi}$ consists of the $n$ most robust and informative SFCs.

[revised manuscript text omitted]

---

## Author Response (AR3)

**3[rd] revision of manuscript**

Journal:      Hydrology and Earth System Sciences (HESS)
Title:        Streamflow characteristics from modelled runoff time series - importance of
              calibration criteria selection
Manuscript:   hess-2016-546
Authors:      Sandra Pool, Marc J. P. Vis, Rodney R. Knight, and Jan Seibert

Dear editor,

We thank you for your and the reviewer's efforts with our manuscript. The reviewer provided again valuable comments on our manuscript, which helped us to further improve the clarity of our manuscript. Below, we reply to each of the comments from the reviewer and indicate the changes that have been done accordingly (marked with blue color). References to pages (P), lines (L), chapters, figures and tables refer to the track-changed revised version of our manuscript. We also did a few additional minor edits that are marked in the track-changed manuscript.

Yours sincerely,
On the behalf of all Co-Authors,
Sandra Pool

**Reviewer 3: comments to 2[nd] revision, response and modifications**

Dear Sjur Kolberg,

Thank you for your efforts with our manuscript. We greatly appreciated the comments, which helped us to improve the manuscript. Please see below our detailed response to each of your comments.

Best regards,

Sandra Pool and Co-Authors

**Major comments**

**Comment 1:** The paper contains no clear definition or distinction between a goodness-of fit measure based on an SFC, and a 'traditional calibration criterion' (TCC). In my view, a comparison of SFCs to TCCs should acknowledge that there may be a transition zone between the two, but still provide a clear distinction which makes the comparison meaningful. SFCs are (citing): 'often used to refer to <such> specific aspects of the flow regime', however, this is also true for $R_{eff}$, $R_{eff}(ln)$, slope of FDC etc. In addition, the analysis use $R_{eff}$ alone as TCC benchmark, although the necessity of combining different TCCs in calibration is well recognised. This constructs a comparison in favour of SFCs. A definition of 'traditional criterion' could be used, for instance along the lines of 'A goodness of fit measure computed from all sim-obs residuals within the calibration period, possibly transformed'. Then distinct groups of SFCs could be recognised by being 'selected from specific seasons or situations', or 'based on duration of events or conditions' etc.

**Reply 1:** We agree that there is no clear boundary between traditional objective functions and SFC based objective functions. We account for this transition by not only calibrating on $R_{eff}$ or SFCs, but also on combinations thereof ($I_{Single\_Reff}$ or $I_{Multi\_Reff}$). The decision to use $R_{eff}$ as benchmark in this study stems from the conclusions of the preceding study (Vis et al., 2015), where $R_{eff}$ outperformed model calibrations with a multi-objective function with respect to SFCs (please see reply to comment 1 in the section 'detailed and specific comments').

In this study we use the term traditional objective function when referring to e.g. Nash-Sutcliffe efficiency or volume error. Such traditional objective functions mostly have a statistical background. In contrast, SFC based objective functions of this study focus on specific runoff aspects with ecological relevance in our study region. We therefore consider them as not being purely statistically

motivated. As suggested by the reviewer, we added a few sentences in the introduction to define traditional and SFC based objective functions.
**Modification:** P4 L10-13

**Comment 2:** With no rules for how SFCs are constructed, statements like 'High flow SFCs tend to be under-estimated' are meaningless. Flashiness index and concentration time both characterise high-flow behaviour, but underestimating one would mean overestimating the other. From table 1 the reader may verify that each SFC used in this paper is scaled so its value is positively related to flow magnitude for the relevant section of the FDC, but this should be asserted in the text when used in conclusions.
**Reply 2:** We agree that drawing conclusions over groups of flow conditions (e.g. high-flow) can lead to misleading conclusions if SFCs of that group represent contrasting hydrograph aspects (as shown in the example given by the reviewer). Fortunately, this is not the case for the flow conditions and SFCs used in this study (please see Table 1 for the definition of SFC). We adapted a sentence in the abstract (P1 L20) and in the results part (P9 L8-10) to make the reader aware of that the results for over- and underestimation only hold for the tested SFCs.
Five out of the 13 tested SFCs are scaled with mean, median or total runoff (Table 1). We agree, that this normalization leads to SFC values that are dependent on the flow magnitude. Although this has an influence on SFC errors, it does not affect the sign of the error. The normalization therefore does not change the conclusions about under- and overestimation of SFCs. We added a few sentences about the normalization of SFCs and its influence on the results in the methods section (P5 L4-7).
**Modification:** P1 L20, P9 L8-10, P5 L4-7

**Comment 3:** Most serious objection: The paper draws conclusions based on subjective evaluations of results. There is no reference to statistical significance or confidence intervals, and very few references to thresholds for 'good simulation', 'small error' etc. In section 2.4.1 on page 5, it is stated that an ensemble of 100 calibrated parameter sets were available for each objective function, allowing analysis of parameter uncertainty. I have not investigated the genetic algorithm used for calibration, but could the variability within this ensemble be used to assess which differences extend beyond mere noise?
**Reply 3:** As suggested by the reviewer, we conducted a parameter uncertainty analysis to estimate the significance of the results (especially the significance of the median value). As null hypothesis we assumed that the probability for over- and underestimation of a SFC is 50%. The two-sided binomial test was used to assess the significance of the results at a confidence level of 0.95. The results are presented in Fig. 9 and section 3.3.
Thresholds for good and poor SFC estimates which were originally introduced in chapter 3.3 were discarded in the adapted manuscript (for a discussion on thresholds please see comment 10 in the section 'detailed and specific comments'). In the results chapters 3.1 and 3.2 we illustrate the difference in model performance related to the five objective functions relative to each other. Since the focus lies on which objective function gives the best SFC estimates, a clear threshold for good or poor simulations is not needed in these two chapters.
**Modification:** P7 L6-8 (methods), P9 L21–30 (results), P31-32 Fig. 9.

**Detailed and specific comments**
**Comment 1:** Page 3, lines 22-24. This sentence is unclear and possibly erroneous. What is meant?
**Reply 1:** We adapted the sentence to make it clearer. The content of the sentence was correct, meaning that in the cited study calibrations with $R_{eff}$ outperformed calibrations with a multi-objective function (consisting of different combinations of $R_{eff}$, $R_{eff}$ of log-transformed flow, volume error and Spearman rank correlation; please see P2 L33-35 and P3 L24-27) with respect to the investigated SFCs. For this reason we selected $R_{eff}$ as benchmark for our study.
**Modification:** P3 L24-26

**Comment 2:** Page 3 line 29 and onwards. Please specify also what is **not** used. For instance in (1) make it clear that this SFC is used alone, in (2) whether or not the multi-SFC vector

includes the SFC being evaluated, and (3) if the TCCs are kept in or excluded when the SFCs are included.

**Reply 2:** Following the research questions we specified in brackets how these objective functions could be defined.

**Modification:** P4 L4-9

**Comment 3:** Page 4 line 13. The reference to NAVD 88 is unnecessary.

**Reply 3:** We removed the reference to NAVD 88.

**Modification:** P4 L23

**Comment 4:** Page 6 line 10: Not yet knowing that the SFCs are normalised to a common scale, the reader may wonder how $R_{eff}$ and an arbitrary SFC can be equally weighted. Just put in a 'normalised (see below)'.

**Reply 4:** Thank you for this suggestion, we added the term 'normalized' to the mentioned sentence.

**Modification:** P6 L24

**Comment 5:** Page 6 line 20: Can you specify how small error are required for a SFC to be robust, and how 'relatively good simulations for other SFC' are required for being informative? These limits have the impact of restricting which SFCs enter the Multi alternative.

**Reply 5:** We agree that a clear threshold value for robustness and information value helps the reader to understand why a particular SFC was selected for $I_{Multi}$. Instead of using a fix threshold value (e.g. normalized model error smaller than 25%), we decided to evaluate the robustness and information value of a SFCs relative to other SFCs (e.g. a potential SFC for $I_{Multi}$ should rank among the SFCs with highest robustness). This enables a more flexible selection of potential SFCs for $I_{Multi}$ with focus on acceptable trade off solutions for all SFCs. We adapted the text accordingly to make our selection procedure clearer.

**Modification:** P6 L30-32, P7 L1-2

**Comment 6:** Sections 3.1.1 and 3.1.2: Would these come more naturally in opposite order? I perceive the $I_{Single}$ experiment as the simplest and most obvious, the one-to-any SFC as more involved.

**Reply 6:** We changed to order of these two catchments as suggested. We therefore also had to change the order of Fig. 3 and Fig. 4 and adapt the figure references in the text accordingly.

**Modification:** P7 section 3.1.1, P8 section 3.1.2, P25-26 (Figures)

**Comment 7:** Would it be informative to summarise in a table for all SFCs what is illustrated in fig 3 for TA1?

**Reply 7:** Fig. 3 shows TA1 estimates for calibrations with each of the 13 $I_{Single}$ objective functions and illustrates that calibrating on a single SFCs often leads to poor estimates of another SFC. This result is not very surprising. Since TA1 can also be considered as a representative example for the other 12 SFCs, we argue that it is not necessary or of major interest to show the results for each of the 13 SFCs.

**Comment 8:** Page 8 line 6. What is required to deserve a 'well simulated' mark? Is there an a priori defined threshold?

**Reply 8:** The term 'well simulated' was not meant in absolute, but rather in relative terms. We adapted the sentence accordingly. Together with the suggested text modification in the methods part (see comment 5) it should become clear that we refer to relative performance of SFCs when evaluating robustness and information value. For the discussion of using well defined values for classifying good and poor SFC estimates please see answer to comment 10.

**Modification:** P8 L20-21

**Comment 9:** Page 8 line 8. Se discussion above about SFC estimates being high or low.

**Reply 9:** Please see answer for comment 10

**Comment 10:** Section 3.3, page 8: This is one of the weaker parts. See the above point on subjective conclusions. The categorisation in lines 19 and 20 are well defined, but then not used for anything. The lowest error class collects everything from perfect match to 10% error, capturing the result for all the mean-flow SFCs, 3 out of 4 low-flow SFCs and 4 out of 5 high-flow SFCs. Still, this paragraph states that all high-flow SFCs and three of four mid-flow SFCs are under-estimated, whereas all except one low-flow SFC were over-estimated' (lines 21-22). Such conclusions need to refer to at least a clearly stated threshold, but preferably to statistical significance.

The SFCs listed as having small vs medium absolute errors in line 23 does not correspond to a sorted grouping of the errors in fig 8. The FH6 error is characterised as 'small', but is larger than the DH16, MA41 and TA1 errors listed as medium, as well as the MA26 and FL2 errors not listed in these two lowest groups. Likewise it is difficult to see from figure 8 why MA26, DH13 and FH7 are identified in line 25 as having large error ranges, while MA41, TA1 or DH16 are not. One gets to suspect that the text is referring to another version of figure 8.

**Reply 10:** We reorganized section 3.3. First, we discarded the categorization of the error magnitudes because they were, as pointed out by the reviewer, to some degree subjective. This made the paragraph about error magnitudes shorter and more concise (P8 L2-10). Secondly, we extended the part of section 3.3 which is about the uncertainty of the over- and underestimation of SFCs (please see comment 3 of major comments).

**Modification:** P8 L2-10

**Comment 11:** The statement in lines 15-17 suggests an identification between goodness of fit and process representation, which this paper neither investigates nor justifies.

**Reply 11:** We agree that we don't explicitly analyze the link between a particular objective function and process representation by e.g. analyzing hydrographs. However, we argue that this link is implicitly present in all model calibration studies. Objective functions are usually chosen to steer calibration in the direction of certain hydrograph aspects. In our case calibration with a particular SFC aims at simulating the hydrological process represented by that SFC. Similarly, calibration with $R_{eff}$ or $I_{Multi}$ aims at simulating many different hydrograph aspects (and thus hydrological processes) simultaneously.

**Comment 12:** Figure 9: The term 'absolute normalised SFC errors' are used. The figure seems to display signed errors.

**Reply 12:** The figure indeed displays the signed errors. We therefore deleted the word "absolute" from the figure capture.

**Modification:** P32 (Fig. 9)

[revised manuscript text omitted]